# RA3 is a reference-guided approach for epigenetic characterization of single cells

Shengquan Chen [1,2,4], Guanao Yan[3,4], Wenyu Zhang[2], Jinzhao Li[2], Rui Jiang[1✉] & Zhixiang Lin [2✉]

The recent advancements in single-cell technologies, including single-cell chromatin accessibility sequencing (scCAS), have enabled profiling the epigenetic landscapes for thousands of individual cells. However, the characteristics of scCAS data, including high dimensionality, high degree of sparsity and high technical variation, make the computational analysis challenging. Reference-guided approaches, which utilize the information in existing datasets, may facilitate the analysis of scCAS data. Here, we present RA3 (Reference-guided Approach for the Analysis of single-cell chromatin Accessibility data), which utilizes the information in massive existing bulk chromatin accessibility and annotated scCAS data. RA3 simultaneously models (1) the shared biological variation among scCAS data and the reference data, and (2) the unique biological variation in scCAS data that identifies distinct subpopulations. We show that RA3 achieves superior performance when used on several scCAS datasets, and on references constructed using various approaches. Altogether, these analyses demonstrate the wide applicability of RA3 in analyzing scCAS data.

[1] Ministry of Education Key Laboratory of Bioinformatics, Bioinformatics Division at the Beijing National Research Center for Information Science and Technology, Center for Synthetic and Systems Biology, Department of Automation, Tsinghua University, Beijing, China. [2] Department of Statistics, The Chinese University of Hong Kong, Hong Kong SAR, China. [3] School of Mathematical Sciences, Zhejiang University, Hangzhou, China. [4]These authors contributed equally: Shengquan Chen, Guanao Yan. ✉email: ruijiang@tsinghua.edu.cn; zhixianglin@cuhk.edu.hk

Chromatin accessibility is a measure of the physical access of nuclear macromolecules to DNA and is essential for understanding the regulatory mechanism[1,2]. For rapid and sensitive probing of chromatin accessibility, assay for transposase-accessible chromatin using sequencing (ATAC-seq) directly inserts sequencing adaptors into accessible chromatin regions using hyperactive Tn5 transposase in vitro[3]. With the recent advancements in technology, single-cell chromatin accessibility sequencing (scCAS) further enables the investigation of epigenomic landscape in individual cells[4,5]. However, the analysis of scCAS data is challenging because of its high dimensionality and high degree of sparsity, as the low copy number (two of a diploid-genome) of DNA leads to only 1–10% capture rate for the hundreds of thousands of possible accessible peaks[6]. The proposed approaches for the analysis of single-cell RNA-seq (scRNA-Seq) data thus present limitations due to the novelty and assay-specific challenges of extreme sparsity and tens of times higher dimensions[6].

Several computational algorithms have been proposed to analyze scCAS data. chromVAR assesses the variation of chromatin accessibility using groups of peaks that share the same functional annotations[7]. scABC calculates weights of cells based on the number of distinct reads within the peak background and then uses weighted $k$-medoids to cluster the cells[8]. cisTopic applies latent Dirichlet allocation model to explore cis-regulatory regions and characterizes cell heterogeneity from the generated regions-by-topics and topics-by-cells probability matrices[9]. Cusanovich et al. proposed a method that performs the term frequency-inverse document frequency transformation (TF-IDF) and singular value decomposition iteratively to get the final feature matrix[5,10]. Scasat uses Jaccard distance to evaluate the dissimilarity of cells and performs multidimensional scaling to generate the final feature matrix[11]. SnapATAC segments the genome into fixed-size bins to build a bins-by-cells binary count matrix and uses principal component analysis (PCA) based on the Jaccard index similarity matrix to obtain the final feature matrix[12]. SCALE combines a variational autoencoder and a Gaussian mixture model to learn latent features of scCAS data[13]. Destin is based on weighted PCA, where the peaks have different weights based on the distances to transcription start sites and the relative frequency of the peaks in ENCODE data[14–16].

Incorporating reference data in analyzing single-cell genomic data can better tackle the high level of noise and technical variation in single-cell genomic data. Most reference-guided methods are designed for single-cell transcriptome data, and they focus primarily on cell type annotation using reference data and marker genes, which limits their application to other downstream analyses, such as data visualization and trajectory inference[17–27]. For the analysis of single-cell chromatin accessibility data, SCATE[28] was recently proposed to reconstruct and recover the "true" chromatin accessibility level for each region in scCAS data utilizing the information in bulk chromatin accessibility data, which is similar to the goal of imputation methods developed for scRNA-Seq data. Massive amounts of bulk chromatin accessibility data have been generated from diverse tissues and cell lines[15,16]. The amount of scCAS data is rapidly increasing[4,5,10,29,30]. Meanwhile, computational tools that collect chromatin accessibility data and efficiently compute chromatin accessibility over the genomic regions facilitate the construction of reference data[31,32].

To utilize the information in existing chromatin accessibility datasets for the analysis of scCAS data, we propose a probabilistic generative model, RA3, in short for Reference-guided Approach for the Analysis of single-cell chromatin Acessibility data. Incorporating reference data built from bulk ATAC-seq data, bulk DNase-seq data, and pseudo-bulk data by aggregating scCAS data, RA3 effectively extracts biological variation in single-cell data for downstream analyses, such as data visualization and clustering. RA3 not only captures the shared biological variation between single-cell chromatin accessibility data and reference data, but also captures the unique biological variation in single-cell data that is not represented in the reference data. RA3 can model known covariates, such as donor labels. Through comprehensive experiments, we show that RA3 consistently outperforms existing methods on datasets generated from different platforms, and of diverse sample sizes and dimensions. In addition, RA3 facilitates trajectory inference and motif enrichment analysis for more biological insight on the cell subpopulations.

## Results

**The RA3 model**. RA3 is a generative model based on the framework of probabilistic PCA[33], and it decomposes the total variation in scCAS data into three components: the component that captures the shared biological variation with reference data, the component that captures the unique biological variation in single cells, and the component that captures other variations (Fig. 1a). More specifically, the first component utilizes the prior information of the projection vectors learned from reference data, and it captures the variation in single-cell data that is shared with the reference data. Choice of the reference data is flexible: it can be the chromatin accessibility profiles of bulk samples or pseudo-bulk samples by aggregating single cells. In practice, the reference data can be incomplete: novel cell types or novel directions of biological variation that are not captured in the reference data can be present in the single-cell data. The second component captures the unique biological variation in single-cell data that is not present in the reference data: it incorporates the spike-and-slab prior to capture the direction of variation that separates a small subset of cells from the other cells, since there may be rare cell types that are not captured in the reference data. The spike-and-slab prior also facilitates RA3 to distinguish biological variation from technical variation, assuming that the direction of variation that separates a small subset of cells more likely represents biological variation. The third component captures the other variations in single-cell data, and it likely represents the technical variation. Other than the three components, RA3 includes another term to model known covariates. The first and second components are used for downstream analyses, and we present results on data visualization, cell clustering, trajectory inference and motif enrichment analysis.

**RA3 decomposes variation in single-cell data**. We first use a simple example as a proof of concept to demonstrate RA3. We collected human hematopoietic cells with donor label BM0828 from a bone marrow scATAC-seq dataset[29] (referred as the human bone marrow dataset). To reduce the noise level, we first adopted a feature selection strategy similar to scABC and SCALE (Methods). We performed the TF-IDF transformation to normalize the scATAC-seq data matrix, implemented PCA, and then performed $t$-distributed stochastic neighbor embedding (t-SNE)[34] to reduce the dimension to two for visualization. This approach (TF-IDF + PCA) is similar to that in Cusanovich2018[5,10], which is among the top three methods suggested in a recent benchmark study[6]. More discussions on TF-IDF transformation are provided in the Methods section. It is hard to separate the majority of the cell types using TF-IDF + PCA, and only CLP and MEP cells are moderately separated from the other cells (Fig. 1b). We then collected a reference data: bulk ATAC-seq samples from four parent nodes in the hematopoietic differentiation tree[29], including samples that correspond to HSC, MPP, LMPP, and CMP cells after fluorescent activated cell sorting. The genomic regions in the

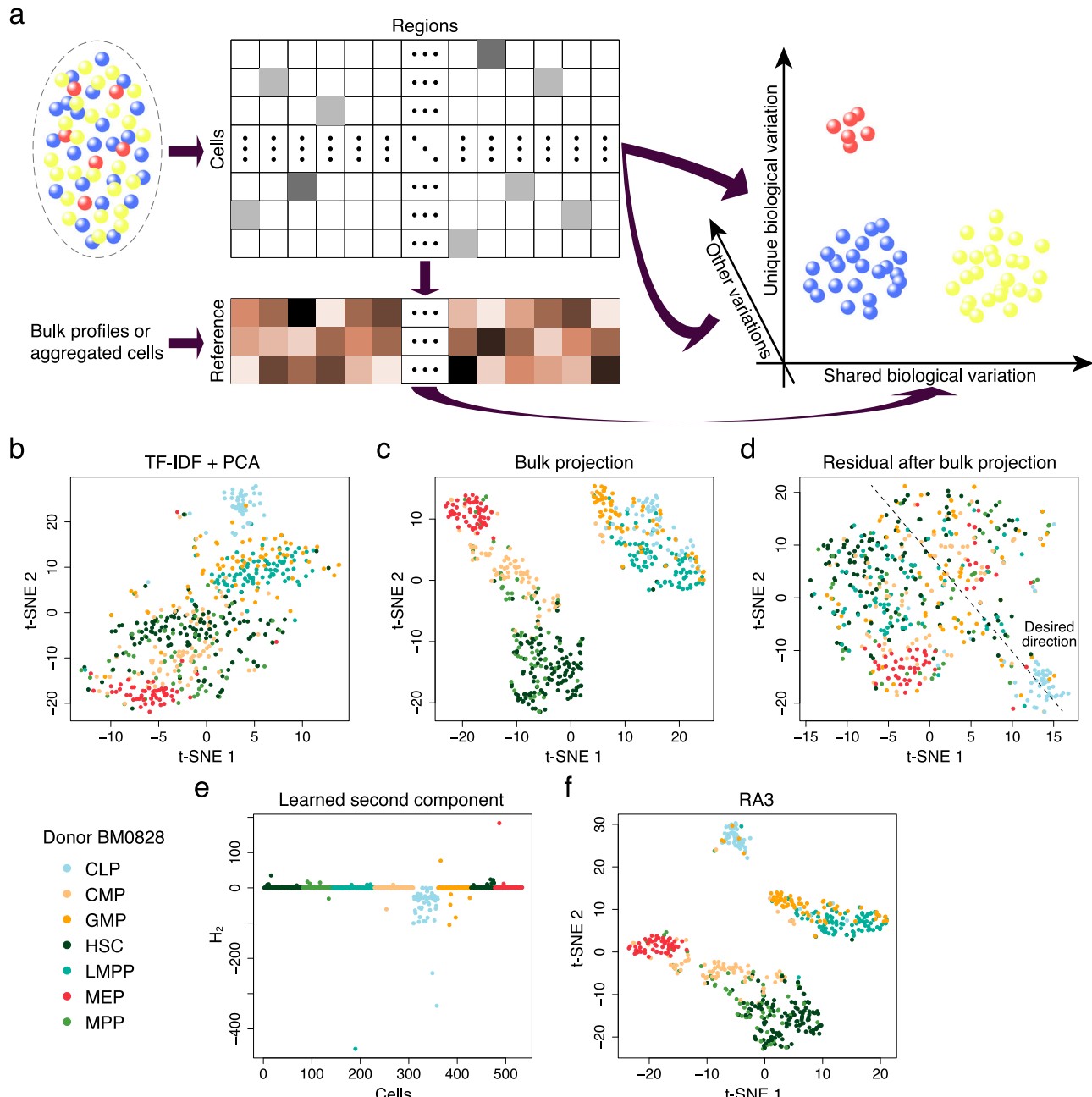

**Fig. 1 The reference-guided approach for the analysis of scCAS data. a** A graphical illustration of the RA3 model. RA3 decomposes the variation in scCAS data into three components: the component that captures the shared biological variation with reference data, the component that captures the unique biological variation in single-cell data, and the component that captures other variations. **b** t-SNE visualization of the cells from donor BM0828 using latent features obtained from TF-IDF + PCA. **c** t-SNE visualization of the cells from donor BM0828 using latent features obtained from bulk projection. **d** We calculated the residuals after the bulk projection. PCA was performed on the residuals, followed by t-SNE visualization. **e** The learned second component with the spike-and-slab prior in RA3. **f** t-SNE visualization using the first two components learned by RA3. TF-IDF term frequency-inverse document frequency transformation, PCA principal component analysis.

reference data are matched with that in the single-cell data. We first considered a simple approach, referred as the bulk projection approach, to utilize the information in the reference data: (1) we applied PCA on bulk ATAC-seq data; (2) we used the projection vectors learned from bulk data to project single-cell data after TF-IDF normalization; (3) for visualization, we applied t-SNE on the projected data to further reduce the dimension to two. This simple approach significantly improves the separation of the cell types (Fig. 1c). A method similar to the above approach was proposed in Buenrostro et al.[29], but the performance of the method was not as good as our simple approach (Supplementary Fig. 1a).

The bulk projection approach implicitly assumes that all the biological variation is shared in single-cell and the reference data, as it only uses the projection vectors learned from the reference data and do not use single-cell data to learn the projection vectors. In this example, using the projection vectors from reference data alone cannot distinguish CLP from the other cells, since the variation of CLP cells is not captured in the reference data. We took residual after the bulk projection, and implemented

PCA + t-SNE on the residual matrix: most cells other than MEP and CLP cells are mixed together, which indicates the presence of strong technical variation in the residual matrix (Fig. 1d). This observation suggests that including the direction of variation learned from single-cell data may help to separate CLP cells, but the direction needs to be carefully chosen because of the strong technical variation. A schematic plot for the desired direction of variation to learn from single-cell data is shown in Fig. 1d: it separates a small subset of cells from the other cells, and the technical variation is weaker along that direction.

Our proposed RA3 models the data after TF-IDF transformation. To overcome the limitation of the bulk projection approach, RA3 not only incorporates prior information from the projection vectors learned from the reference data as the first component in the decomposition of variation, but also incorporates a second component to model the unique biological variation in single-cell data. The key for the second component in RA3 is a spike-and-slab prior[35,36], which facilitates the model to detect directions that lead to good separation of a small number of cells from the other cells, and not necessarily directions with the largest variation. Using the direction of the largest variation can be problematic given that the technical variation can be strong. Applying RA3 to the example, the second component with the spike-and-slab prior successfully distinguishes CLP cells (Fig. 1e). We note that the labels for CLP cells are not used in RA3 to detect the direction that separates CLP cells. In this example, RA3 effectively utilizes the prior information in reference data to separate CMP, GMP, HSC/MPP, LMPP, and MEP cells, and it also captured the unique variation in single-cell data, which separates CLP cells from GMP and LMPP cells (Fig. 1f). The loadings of the second component in RA3 provide functional insight on the cell subpopulations. Using the top 1000 peaks with largest magnitude in the loadings of the second component (we focus on peaks with negative loadings as the sign of $H_2$ for the identified cell subpopulation is mostly negative), we performed Genomic Region Enrichment of Annotation Tool (GREAT)[37] analysis to identify significant pathways associated with the identified cell subpopulation in the second component (Methods). The top five pathways with smallest $p$ values from the binomial test are regulation of lymphocyte activation, Fc receptor signaling pathway, immune response-regulating cell surface receptor signaling pathway, immune response-activating cell surface receptor signaling pathway, and positive regulation of lymphocyte activation (see Supplementary Table 1 for the complete enrichment results). These enriched pathways are consistent with the function of CLP cells: CLP cells serve as the earliest lymphoid progenitor cells and give rise to T-lineage cells, B-lineage cells, and natural killer (NK) cells. To summarize, the second component in RA3 not only identifies the rare cell subpopulation, but also provides functional insight of the identified cell subpopulation.

**RA3 builds effective reference from massive bulk data**. The previous example of hematopoietic cells utilizes reference data constructed from manually curated bulk samples that have relevant biological context with the single-cell data. The cellular composition in single-cell data is generally unknown. It can be desirable to utilize bulk reference data generated from diverse biological contexts and cell types, such as all the bulk chromatin accessibility data generated in the ENCODE project[15,16]. The implementation of RA3 requires matched regions/features in the target single-cell data and the reference data. The web-based tool OPENANNO[31] provides a convenient way to construct the reference data: the input for OPENANNO is the peak information in single-cell data, and OPENANNO will calculate the accessibility of these peaks in 871 bulk DNase-seq samples of

diverse biological context collected from ENCODE, which can be used as the reference data. Note that this approach requires the BAM files for the reference samples to calculate accessibility, an alternative approach that does not require BAM files will be discussed later. With the peak information in the human bone marrow dataset[29], we used OPENANNO to construct the reference data with samples of diverse biological context. The reference data constructed in this way achieved similar performance as the manually curated reference data using only the relevant cell types (Supplementary Fig. 1b).

We next applied RA3 to three other single-cell datasets: (1) a mixture of human GM12878 and HEK293T cells (referred as the GM/HEK dataset)[5]; (2) a mixture of human GM12878 and HL-60 cells (referred as the GM/HL dataset)[5]; (3) an in silico mixture of H1, K562, GM12878, TF-1, HL-60 and BJ cells (referred as the InSilico mixture dataset)[4]. We first manually constructed the reference using BAM files of bulk DNase-seq samples from the relevant cell lines (Methods). The performance significantly improved over the approach not using reference data (Fig. 2a,b and Supplementary Fig. 1d, e), which indicates that the abundant BAM files of bulk chromatin accessibility profiles in the literature can be fully used to help the analysis of scCAS data. When we constructed the reference data with OPENANNO using all the 871 bulk samples (Methods), the performance of RA3 is comparable with the implementation using only the relevant cell lines for reference (Fig. 2c and Supplementary Fig. 1d, e).

We also considered an alternative approach to construct the reference using only the peak files (BED files) of the reference samples, to address the situation when BAM files are not available. The reference data can be constructed by counting the number of peaks in reference sample that overlap with every peak in single-cell data (Methods). This approach to construct the reference data also led to satisfactory performance (Fig. 2d and Supplementary Fig. 1c–e), which suggests that we can build useful reference with the peak files of bulk data when BAM files are not available. Therefore, databases that collect more comprehensive biological samples but only provide the peak information for each bulk sample, such as Cistrome DB[32], may further facilitate the usage of RA3.

**RA3 incorporates pseudo-bulk data as reference**. It can be difficult to obtain the bulk samples for certain cell populations, especially for the cells in frozen or fixed tissues, where cell sorting is challenging to implement. The recent efforts of cell atlas consortiums have generated massive amounts of single-cell transcriptome data for whole organisms[38–47], and single-cell chromatin accessibility data are rapidly increasing[4,5,10,29,30]. We can construct pseudo-bulk reference data by aggregating single cells of the same type/cluster to alleviate the high degree of sparsity in scCAS data. As a proof of concept, we first look at a single-nucleus ATAC-seq dataset generated from mouse forebrain (referred as the mouse forebrain dataset)[30], where the cell type labels were provided, including astrocyte (AC), three subtypes of excitatory neuron (EX1, EX2, and EX3), two subtypes of inhibitory neuron (IN1 and IN2), microglia (MG), and oligodendrocyte (OC). We randomly split the cells in this dataset into half: half of the cells were aggregated by the cell types to build the pseudo-bulk reference, and the other half of the cells were used as the single-cell data. It is hard to separate the subtypes of excitatory neurons using TF-IDF + PCA (Supplementary Fig. 2a). RA3 using the pseudo-bulk reference successfully identified all the cell types, with moderate separation in the three subtypes of excitatory neurons, EX1, EX2, and EX3 (Fig. 2e). To investigate the influence of incomplete reference data, we left out MG and OC cells in constructing the pseudo-bulk reference data. As expected,

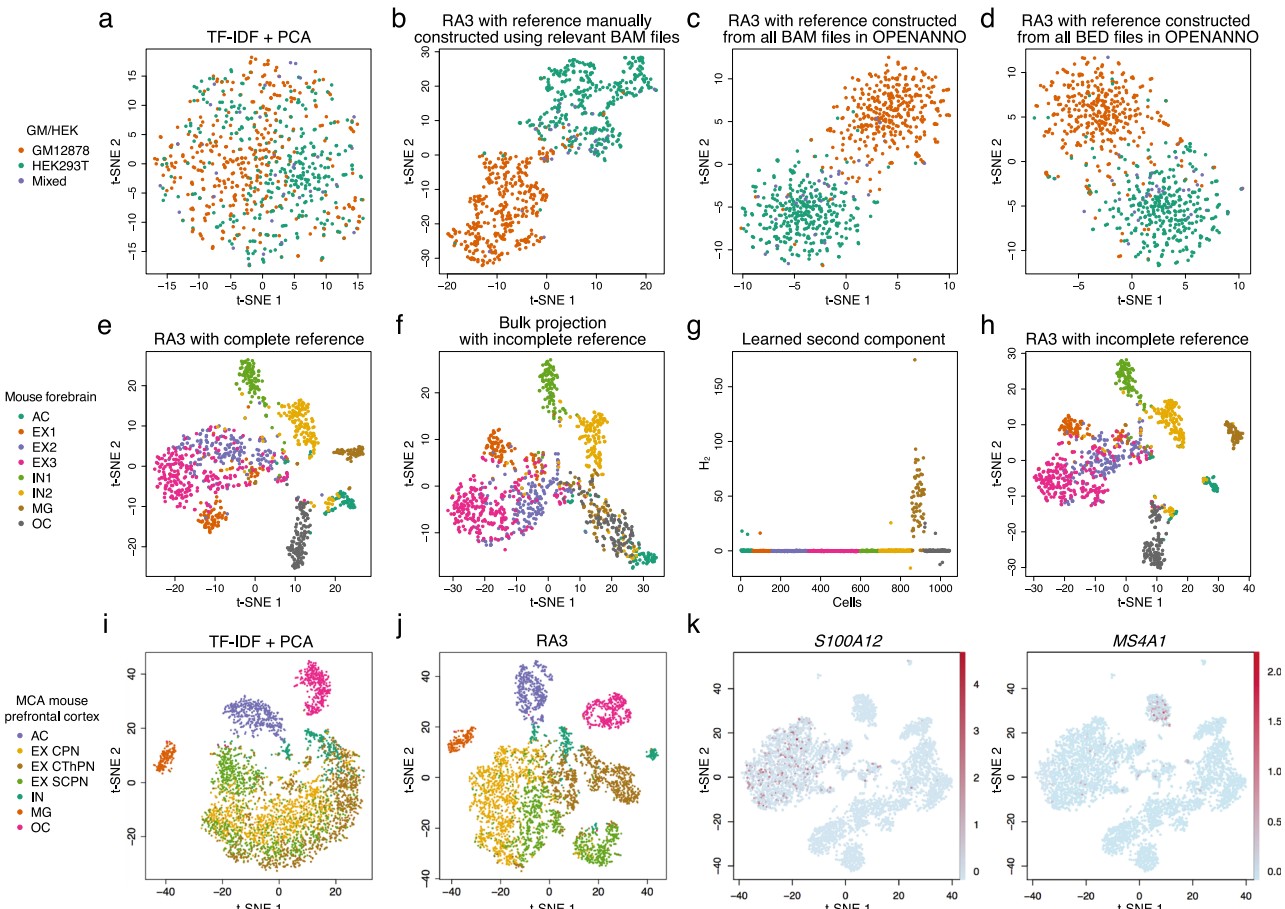

**Fig. 2 RA3 incorporates reference data constructed from different sources.** t-SNE visualizations of the cells in the GM/HEK dataset using latent features obtained from **a** TF-IDF + PCA, and from RA3 using reference data constructed from different samples, including **b** BAM files of bulk GM12878 and HEK293T DNase-seq samples, **c** BAM files of all the bulk samples in OPENANNO, and **d** BED files of all the bulk samples in OPENANNO. **e** We split the cells in the mouse forebrain dataset into half: half of the cells were used to construct pseudo-bulk reference, and the other half were treated as single-cell data. t-SNE visualization using the latent features learned by RA3 with the complete reference is shown. **f** We also constructed an incomplete pseudo-bulk reference by leaving out MG and OC cells. t-SNE visualization using the latent features obtained by bulk projection with incomplete reference is shown. We implemented RA3 with the incomplete reference: **g** the learned second component with spike-and-slab prior and **h** t-SNE visualization of the learned latent features are shown. **i** t-SNE visualizations of cells in the mouse prefrontal cortex dataset, using the latent features obtained from TF-IDF + PCA and **j** the latent features obtained from RA3 with pseudo-bulk reference constructed from the mouse forebrain dataset. **k** t-SNE visualizations of cells in the 10X PBMC dataset using the latent features obtained from RA3 with pseudo-bulk reference constructed from another PBMC dataset. Chromatin accessibility of *S100A12* (a marker gene of monocytes) and *MS4A1* (a marker gene of B cells) is projected onto the visualizations, respectively. TF-IDF term frequency-inverse document frequency transformation, PCA principal component analysis.

the bulk projection approach cannot distinguish MG and OC cells (Fig. 2f). The spike-and-slab prior in RA3 successfully detected the directions that lead to good separation of MG and OC cells (Fig. 2g and Supplementary Fig. 2b), and RA3 led to improved separation of all the cell types (Fig. 2h). We also performed an experiment where the overlap between scCAS data and the reference data gradually decreases. We gradually and randomly downsampled the cells of the shared cell types (AC, EX1, EX2, EX3, IN1, and IN2) in scCAS data, and the cell types (MG and OC) unique to scCAS data remain unchanged. Before downsampling, MG and OC constitute 18.1% of the total cells in scCAS data. At the end point of downsampling, only 10.0% cells of the shared cell types are retained, and MG and OC constitute 68.7% of the total cells in scCAS data. We used the default parameters in RA3. As shown in Supplementary Fig. 3, RA3 successfully separated MG and OC cells even when they become the majority cell types in scCAS data.

To further demonstrate the advantage of RA3 for the analysis of single-cell epigenetic profiles utilizing existing single-cell data,

we collected single cells from a sciATAC-seq dataset of the adult mouse brain (referred as the MCA mouse brain dataset)[10]. We used the provided cell type labels to evaluate different methods[10]. We first look at cells from the mouse prefrontal cortex. It has been suggested that there is heterogeneity within excitatory neurons in mouse prefrontal cortex[10]. TF-IDF + PCA can hardly distinguish the subtypes of excitatory neurons and inhibitory neurons (Fig. 2i). We used the complete mouse forebrain dataset[30] to construct a pseudo-bulk reference for RA3 (Methods). RA3 achieved much better separation for the subtypes of excitatory neurons (Fig. 2j). RA3 also achieved better performance on three other brain tissues in the MCA mouse brain dataset, including cerebellum and two replicates of the whole brain (Supplementary Fig. 2d–f).

Next, we collected peripheral blood mononuclear cells (PBMCs) from 10X Genomics (referred as the 10X PBMC dataset). The labels of cell type are not provided in this dataset, and eight cell populations inferred by cell markers are suggested in recent studies[6,9,48], including CD34+ cells, NK cells, dendritic

cells, monocytes, lymphocyte B cells, lymphocyte T cells, and terminally differentiated CD4 and CD8 cells. For the implementation of RA3, we used a previously published PBMC dataset[49] to construct the pseudo-bulk reference (Methods). We first performed dimension reduction with RA3 and then performed clustering on the low-dimensional representation. To evaluate the performance, we adopted the Residual Average Gini Index (RAGI) score[6], which calculates the difference between (a) the variability of marker gene accessibility across clusters and (b) the variability of housekeeping gene accessibility across clusters, and a larger RAGI score indicates a better separation of the clusters (Methods). RA3 achieved a RAGI score of 0.152, while TF-IDF + PCA achieved a RAGI score of 0.110. Aside from RAGI score, RA3 also led to more compact patterns for marker gene activity (Fig. 2k and Supplementary Fig. 2c), compared with TF-IDF + PCA (Supplementary Fig. 2g). In the previous implementations of RA3, we used the peaks in the target single-cell data to calculate accessibility for the reference data. When we use the peaks in the reference data to calculate accessibility for the target single-cell data, the RAGI score for RA3 is 0.150. This implementation demonstrates that RA3 can be implemented when only the count matrix of the reference data is available.

It may be challenging to reliably detect the peaks for the rare cell subpopulations in the target scCAS data. We present two approaches to tackle this issue. (1) The first is to use peaks identified in the reference data. In the 10X PBMC scCAS dataset, we have shown that the clustering performance is similar when we use the peaks in the target scCAS data or the peaks in the reference data: RAGI scores are 0.152 (scCAS peaks) vs 0.150 (reference peaks). The cell atlas consortiums will generate large amounts of data that can be used as reference. We expect that the peak annotations in reference data that RA3 can take advantage of will become increasingly comprehensive in the future. So the first approach will become appealing to tackle the issue of peak calling for rare subpopulations when reference data become more abundant. (2) Our second approach is to iterate between peak calling and clustering with RA3. The peaks may not be reliably detected at the first place. After implementing RA3 and clustering, there will be moderate separation of the cell types. Then we redo peak calling within each obtained cell cluster, and reimplement RA3 on these peaks. We expect that iterations between peak calling and clustering with RA3 will improve peak detection and identification of the rare cell subpopulations. As a proof of concept, we implemented this procedure on the mouse forebrain dataset. We first identified cell type-specific peaks by the hypothesis testing procedure in scABC, and held out the top 500 peaks with minimum $p$ values for the MG and OC cell types, respectively. Note that MG and OC cells are not present in the reference data. We used the default parameters in RA3. RA3 cannot distinguish MG and OC cells without the MG- and OC-specific peaks (Supplementary Fig. 4a). We then performed cell clustering with RA3 + Louvain clustering, performed peak calling within each obtained cell cluster (we followed the pipeline of peak calling provided by the original study of the mouse forebrain data[30]), and merged these peaks to obtain new features. Using these new features, RA3 successfully identified MG and OC cells (Supplementary Fig. 4b), which demonstrates the effectiveness of the iterative peak calling strategy. We note that the first approach and the second approach can be combined, i.e., using both reference peaks and de novo peaks in scCAS data.

To summarize, the examples implementing RA3 with pseudo-bulk reference data demonstrate the potential that RA3 can utilize existing scCAS data to facilitate the analysis of newly generated scCAS data.

**Comparison with other methods**. RA3 was benchmarked against six baseline methods, including scABC[8], Cusanovich2018[5,10], Scasat[11], cisTopic[9], SCALE[13], and SnapATAC[12] (Methods). We evaluated the methods by dimension reduction and clustering with the provided cell labels (except for the 10X PBMC dataset, where the cell labels are not provided): We implemented t-SNE and uniform manifold approximation and projection (UMAP)[50] to further reduce the low-dimensional representation provided by each method to two for visualization. To evaluate the clustering performance, we implemented Louvain clustering on the low-dimensional representation provided by each method as suggested by Chen et al.[6], and we assessed the clustering performance by adjusted mutual information (AMI), adjusted Rand index (ARI), homogeneity score (homogeneity), and normalized mutual information (NMI). For the 10X PBMC dataset, we used RAGI score to evaluate the clustering performance since the cell labels are not available. We only evaluated the clustering performance for scABC because it does not perform dimension reduction.

We first evaluated the performance on the human bone marrow dataset[29]. We used all the bulk chromatin accessibility data provided by Buenrostro et al.[29] as the reference data in RA3. To evaluate different methods, we used both the subset of cells and all the cells in this dataset.

(1) CLP, LMPP and MPP cells. These cells come from two donors, BM0828 and BM1077. Scasat and Cusanovich2018 cannot distinguish MPP and LMPP (Fig. 3a). Although SCALE, cisTopic, and SnapATAC can separate the three cell types, the effect of the two donors is obvious (Fig. 3a and Supplementary Fig. 5), which leads to poor clustering performance for the three cell types (Fig. 4a and Supplementary Fig. 6). When the variation of different donors is not of interest, it is necessary to incorporate donor labels as covariates[51], while all the baseline methods do not include such a component. RA3 benefited from using reference data and from including donor labels as the covariates, and outperformed the baseline methods in both visualization (Fig. 3a) and clustering (Fig. 4a and Supplementary Fig. 6).

(2) Cells from donor BM0828. Scasat and Cusanovich2018 cannot separate most cell types except CLP and MEP (Fig. 3b). We encountered an error message with output "Nan" when implementing SCALE with the default parameters in this dataset. So we did not include SCALE in this comparison. cisTopic and SnapATAC performed reasonably well except for separating CMP cells (Fig. 3b). RA3 performed the best in both visualization (Fig. 3b) and clustering (Fig. 4a and Supplementary Fig. 6).

(3) The full dataset with all the cells and donors. Although cisTopic, Cusanovich2018, and SnapATAC achieved reasonable separation of the cells, their performance was influenced by the effect of donors, especially for HSC, MPP, and LMPP cells (Fig. 3c). RA3 achieved the best clustering performance (Fig. 4a and Supplementary Fig. 6).

For the three datasets with mixture of cell lines[4,5], we constructed the reference in RA3 using bulk DNase-seq samples with relevant biological context for the target single-cell data. Compared to the baseline methods, RA3 also achieved satisfactory performance on the GM/HEK, GM/HL, and InSilico mixture datasets (Supplementary Fig. 7).

We then assessed the performance on the mouse forebrain dataset[30]. We randomly split the cells in this dataset into half: half of the cells were aggregated by the provided cell type labels to build the pseudo-bulk reference, and the other cells were used as the single-cell data. All the baseline methods can hardly distinguish the three subtypes of excitatory neuron (EX1, EX2, and EX3), while RA3 can separate EX1 and it achieved moderate separation of EX2 and EX3 (Figs. 3d and 4a and Supplementary Fig. 6). To mimic platforms that generate sparser scCAS data, we downsampled the reads in single-cell data (Methods). RA3

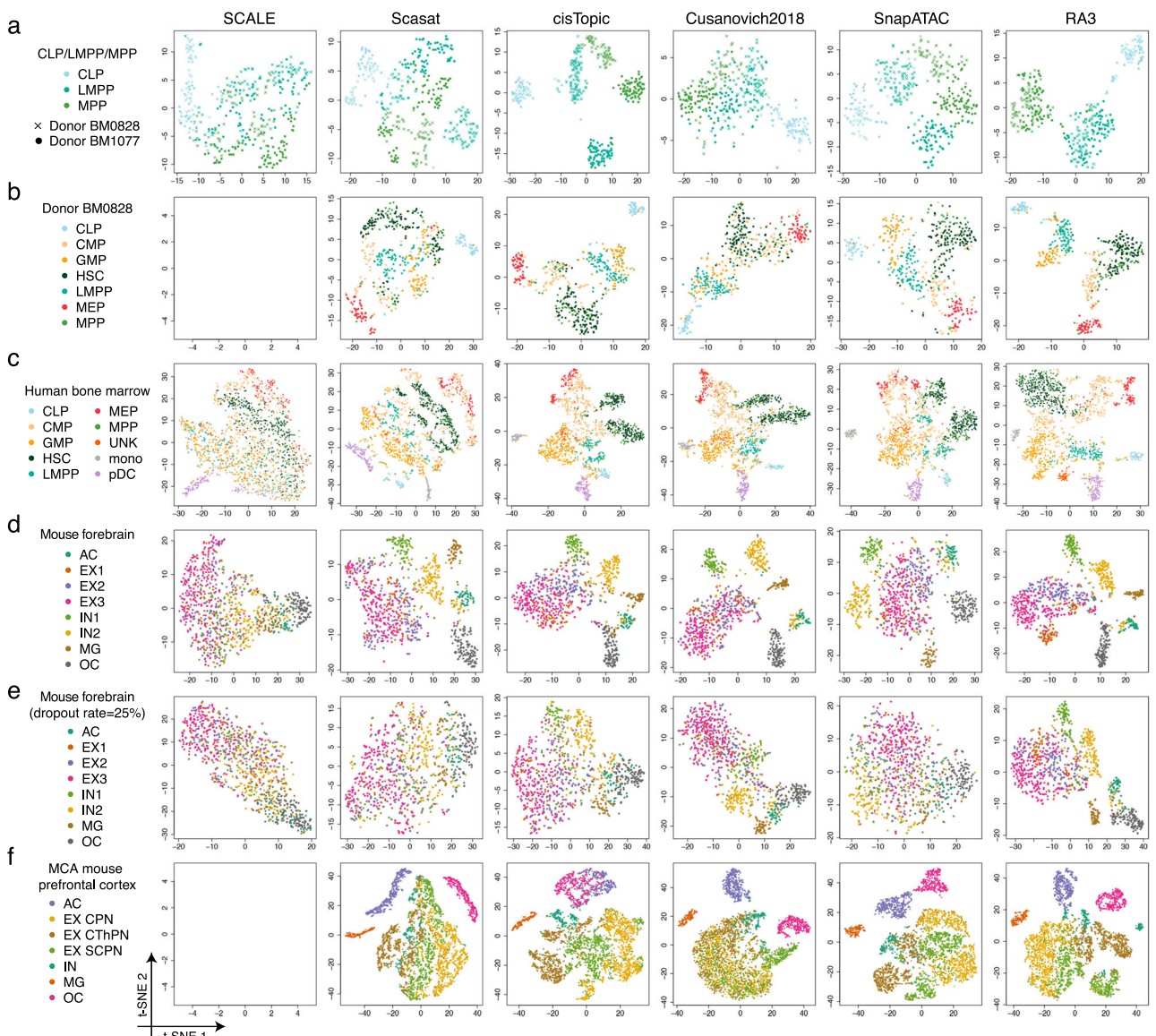

**Fig. 3 Evaluation of the visualization of scCAS data. a** The dataset of CLP/LMPP/MPP cells. **b** The dataset of donor BM0828. **c** The human bone marrow dataset. **d** The mouse forebrain dataset (half). **e** The mouse forebrain dataset (half) with 25% dropout rate. **f** The dataset of mouse prefrontal cortex. For all the datasets, we obtained the latent features from SCALE, Scasat, cisTopic, Cusanovich2018, SnapATAC, and RA3, and then implemented t-SNE for visualization.

consistently outperformed other methods when the dropout rate varies from 5 to 50% (Fig. 4b and Supplementary Fig. 8). Compared with the baseline methods, RA3 was less affected when the dropout rate increases, and RA3 still achieved reasonable separation of the cells when dropout rate = 25% (Fig. 3e). This observation suggests the increased benefit of utilizing reference data when the single-cell dataset has higher degree of sparsity.

We also evaluated the performance on the MCA mouse brain dataset[10]. We used the complete mouse forebrain dataset[30] to construct a pseudo-bulk reference. We first look at cells from the mouse prefrontal cortex. Cusanovich2018 cannot distinguish excitatory neurons as previously reported[10] (Fig. 3f). We encountered an error when implementing SCALE on this dataset. Scasat achieved a slight improvement over Cusanovich2018 in separating excitatory neurons. cisTopic, SnapATAC, and RA3 all provided better separation of the inhibitory neurons and the subtypes of excitatory neurons (Fig. 3f). RA3 achieved better

separation of ACs and OCs (Fig. 3f), and the best overall clustering performance (Fig. 4a). RA3 also outperformed the baseline methods on three other datasets, including mouse cerebellum and two samples of the mouse whole brain (Supplementary Fig. 7).

We finally tested the performance on the 10X PBMC dataset. We used the previously published dataset of PBMC cells[49] to construct a pseudo-bulk reference based on the provided cell type labels. RA3 consistently outperforms the baseline methods according to the RAGI score (Fig. 4c). In addition, RA3 led to more compact patterns for marker gene activity (Supplementary Fig. 9).

In all the examples, the results of UMAP visualization are similar to that of t-SNE visualization (Supplementary Fig. 5). We further evaluated the stability of clustering performance by implementing bootstrap on the datasets in Fig. 4a. To be more specific, we generated ten bootstrap samples by random sampling

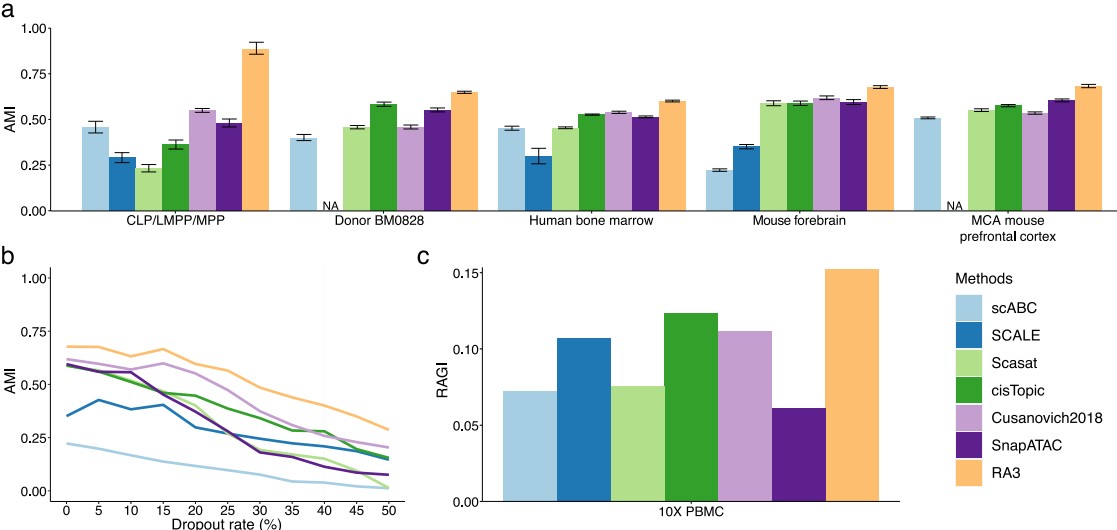

**Fig. 4 Assessment of the clustering results.** We implemented Louvain clustering on the low-dimensional representation provided by each method to get the cluster assignments. The cluster assignments for scABC were obtained directly from the model output. **a** The clustering performance using different methods evaluated by adjusted mutual information (AMI). The measure of center for the error bars denotes the AMI for different methods. The error bar denotes the estimated standard error in ten bootstrap samples. **b** The clustering performance using different methods on the mouse forebrain dataset (half) at different dropout rates evaluated by AMI. **c** The clustering performance using different methods on the 10X PBMC dataset evaluated by Residual Average Gini Index (RAGI) score.

with replacement for each of the dataset. We then performed cell clustering on the bootstrap samples using different methods. The error bars in Fig. 4a represent the standard errors estimated by bootstrap. In addition to the indexes of clustering performance, we present the clustering tables (Supplementary Fig. 10). RA3 achieved a cleaner separation of the cell types, as the clustering counts are more concentrated on the diagonals of the clustering tables. To assess the computational efficiency and scalability of RA3, we benchmarked the running time usage on datasets of different sizes. As shown in Supplementary Fig. 11a, RA3 can scale well with the size of scCAS dataset, especially for large datasets. The computational time of RA3 on datasets with ~5K, 10K, 20K, and 50K cells are 1.7, 7.5, 16.5, and 17.5 min, respectively. The size of the target scCAS dataset is dominant in determining the computational time of RA3, and the number of components learned from the reference data has a relatively minor effect on the computational time (Supplementary Fig. 11b).

**RA3 facilitates trajectory inference and motif analysis**. Other than data visualization and cell clustering, the output of RA3 can also be implemented in downstream analyses including trajectory inference and motif enrichment analysis. We use the cells from donor BM0828 for illustration. We implemented Slingshot[52] for the trajectory inference, which is suggested in a benchmark study[53]. The inputs for Slingshot include the low-dimensional representation provided by RA3 and the cluster labels obtained from RA3 + Louvain clustering, and the output for Slingshot is the smooth curves representing the estimated cell lineages (Methods). RA3 with Slingshot revealed the differentiation lineage and the inferred trajectory greatly mimics the hematopoietic differentiation tree[29] (Fig. 5a). We then performed motif enrichment analysis for the clusters identified by RA3 + Louvain clustering. Cluster-specific peaks are needed for motif analysis and we identified them by the hypothesis testing procedure in scABC using the raw read counts and the cluster labels provided by RA3 + Louvain clustering[8]. Note that for cell-level analysis (visualization, clustering, and trajectory inference), we used TF-

IDF transformed data as the input; for peak-level analysis (differential peak analysis), we used the raw read counts as part of the input. We selected the top 1000 peaks with the smallest $p$ values for each cluster, and then applied chromVAR[7] to infer the enriched transcription factor (TF) binding motifs within these peaks (Methods). Visualization of the top 50 most variable TF binding motifs is shown (Fig. 5b). Thirty-nine of the top 50 most variable TF binding motifs have been implicated in hematopoietic development (Supplementary Table 2). Among them, some TF binding motifs are specific to one or two clusters (Fig. 5b), and previous literature further corroborates the role of these TFs/motifs in the same cell types that the clusters represent: *EBF1*, *TCF3*, and *TCF4* are specific to cluster 7, which corresponds to CLP cells[29,54,55]; *JDP2*, *CEBPB*, and *CEBPE* are specific to cluster 6, which corresponds to GMP cells[29,56,57]; and *GATA1::TAL1*, *GATA2*, and *GATA3* are specific to cluster 4, which corresponds to MEP cells[58,59].

## Discussion
In this work, we have developed RA3 for the analysis of high-dimensional and sparse single-cell epigenetic data using a reference-guided approach. RA3 simultaneously models the shared biological variation with reference data and the unique variation in single-cell data that identifies distinct cell subpopulations by incorporating the spike-and-slab prior. We have shown that the reference data in RA3 can be constructed from various sources, including bulk ATAC-seq, bulk DNase-seq, and pseudo-bulk chromatin accessibility data, which will facilitate the usage of RA3. We demonstrated that RA3 outperforms baseline methods in effectively extracting latent features of single cells for downstream analyses, including data visualization and cell clustering. RA3 also facilitates trajectory inference and motif enrichment analysis for scCAS data. In addition, RA3 is robust and scalable to datasets generated with different profiling technologies, and of diverse sample sizes and dimensions.

Finally, our modeling framework is flexible and can be extended easily. We can incorporate other types of single-cell profiles

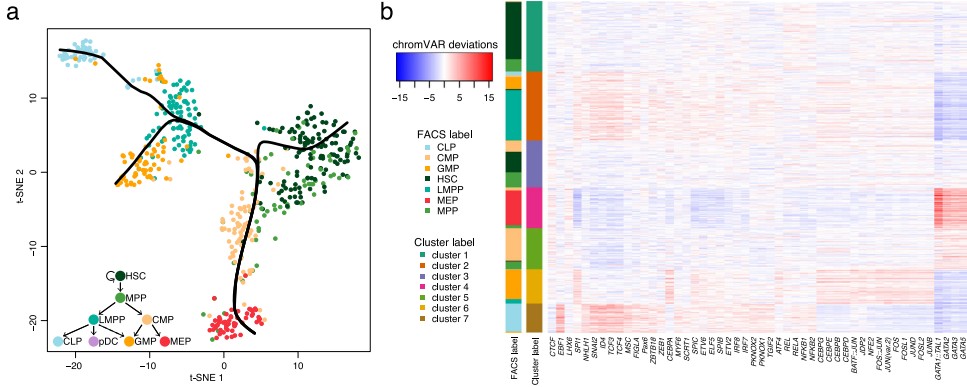

**Fig. 5 Trajectory inference and motif enrichment analysis. a** t-SNE visualization of the cells from donor BM0828 and the inferred trajectory with Slingshot using the output of RA3 and Louvain clustering. The hematopoietic differentiation tree[29] is shown on the bottomleft. **b** The top 50 most variable TF binding motifs within the cluster-specific peaks for the cells of donor BM0828. The deviations calculated by chromVAR are shown. FACS fluorescent activated cell sorting.

as reference data to avoid the situation that the cell types in scCAS data have not been adequately studied from the epigenetic landscape. Besides, we can also extend our approach to non-linear projection of scCAS data by incorporating deep neural networks to capture higher level features.

## Methods

**The model of RA3.** We first apply TF-IDF transformation to the read count matrix in scCAS data. TF-IDF transformation achieves two goals in analyzing scCAS data: (1) it normalizes for sequencing depth; (2) it upweights peaks/regions that do not occur very frequently and downweights prevalent peaks/regions. The peaks/regions that are less frequent in scCAS data tend to represent features that distinguish the cell types, and giving these features higher weights improves separation of the cell types (Supplementary Fig. 12). RA3 decomposes the variation in the scCAS data after TF-IDF transformation into three components: the shared biological variation with the reference data, the unique biological variation in scCAS data, and other variations. Aside from the three components, RA3 also includes a term for known covariates.

Consider the random vector $\mathbf{y}_j \in \mathbb{R}^{p \times 1}$, which represents the observed $p$ features/regions for the $j$th cell among $n$ cells in total. Borrowing the general framework of probabilistic PCA[33], RA3 models $\mathbf{y}_j$ as the following:

$$\mathbf{y}_j | \boldsymbol{\lambda}_j \sim \mathcal{N}_p(\boldsymbol{\lambda}_j, \sigma^2 \mathbf{I}_p), \tag{1}$$

$$\boldsymbol{\lambda}_j = \boldsymbol{\beta}\mathbf{x}_j + \mathbf{W}\mathbf{h}_j, \boldsymbol{\lambda}_j \in \mathbb{R}^{p \times 1}, \tag{2}$$

where $\mathbf{x}_j \in \mathbb{R}^{q \times 1}$ represents $q$ known covariates, including the intercept and other variables, such as the labels of donors or batches, and $\boldsymbol{\beta} \in \mathbb{R}^{p \times q}$ represents the unknown coefficients for $\mathbf{x}_j$. The variables $\mathbf{W} \in \mathbb{R}^{p \times K}$ and $\mathbf{h}_j \in \mathbb{R}^{K \times 1}$ are latent variables: the columns in $\mathbf{W}$ have similar interpretation as the projection vectors in PCA, and $\mathbf{h}_j$ can be interpreted as the low-dimensional representation of $\mathbf{y}_j$. We further decomposes the term $\mathbf{W}\mathbf{h}_j$ into three components:

$$\mathbf{W}\mathbf{h}_j = \mathbf{W}_1 \mathbf{h}_{j_1} + \mathbf{W}_2 \mathbf{h}_{j_2} + \mathbf{W}_3 \mathbf{h}_{j_3}, \tag{3}$$

where the dimensions are $\mathbf{W}_1 \in \mathbb{R}^{p \times K_1}$, $\mathbf{W}_2 \in \mathbb{R}^{p \times K_2}$, $\mathbf{W}_3 \in \mathbb{R}^{p \times K_3}$, $\mathbf{h}_{j_1} \in \mathbb{R}^{K_1 \times 1}$, $\mathbf{h}_{j_2} \in \mathbb{R}^{K_2 \times 1}$, $\mathbf{h}_{j_3} \in \mathbb{R}^{K_3 \times 1}$. So we have $\mathbf{W} = [\mathbf{W}_1 \ \mathbf{W}_2 \ \mathbf{W}_3]$ and $\mathbf{h}_j = [\mathbf{h}_{j_1}^T \ \mathbf{h}_{j_2}^T \ \mathbf{h}_{j_3}^T]^T = [\mathbf{h}_{j_1}^T \ \mathbf{h}_j^{*T}]^T$ with $\mathbf{h}_j^* \triangleq [\mathbf{h}_{j_2}^T \ \mathbf{h}_{j_3}^T]^T$.

We set $\mathbf{W}_1$ to be equal to the $K_1$ projection vectors learned from PCA on the reference data, so the first component $\mathbf{W}_1 \mathbf{h}_{j_1}$ utilizes prior information from the reference data and captures the shared biological variation among scCAS data and the reference data.

For the columns in $\mathbf{W}_2$ and $\mathbf{W}_3$, the prior specification is similar to that in Bayesian PCA[60]:

$$\mathbf{w}_k \sim \mathcal{N}_p(0, \alpha_k^{-1}\mathbf{I}_p), \text{ for } k = K_1 + 1, \ldots, K, \tag{4}$$

where hyper-parameters $\boldsymbol{\alpha} = \{\alpha_{K_1+1}, \ldots, \alpha_K\}$ are precision parameters that controls the inverse variance of the corresponding $\mathbf{w}_k$, for $k = K_1 + 1, \ldots, K$.

Let $\boldsymbol{\gamma}_j = [\gamma_{1j}, \ldots, \gamma_{Kj}]^T \in \mathbb{R}^{K \times 1}$ denote a $K$-dimensional binary latent vector for cell $j$. We assume the following priors for $\boldsymbol{\gamma}_j$ and $\mathbf{h}_j$:

$$\begin{cases} \gamma_{kj} = 1, h_{kj}|\gamma_{kj} \sim \gamma_{kj}\mathcal{N}(0,1), & k = 1, \ldots, K_1, K_1 + K_2 + 1, \ldots, K \\ \gamma_{kj} \sim \text{Bernoulli}(\theta), h_{kj}|\gamma_{kj} \sim (1-\gamma_{kj})\mathcal{N}(0,\tau_0^2) + \gamma_{kj}\mathcal{N}(0,\tau_1^2), & k = K_1 + 1, \ldots, K_1 + K_2 \end{cases}$$
$$\tag{5}$$

Note that for the entries in $\mathbf{h}_{j_1}$ and $\mathbf{h}_{j_3}$, the prior specification is the same as that in Bayesian PCA, while for the entries in $\mathbf{h}_{j_2}$, we assume the spike-and-slab[35,36] prior with pre-specified parameters $\tau_0 < 1$ and $\tau_1 > 1$. The spike-and-slab prior is the key that encourages the model to find directions in $\mathbf{W}_2$ that separate rare and distinct cell types from the other cells, and it will help us to distinguish biological variation from technical variation. Therefore, the second component $\mathbf{W}_2 \mathbf{h}_{j_2}$ captures the variation unique in scCAS data that separates distinct and rare cell types from the other cells. We note that the sparsity specification on $\mathbf{h}_j$ is different from that in sparse PCA[61], where $\mathbf{W}$ is assumed to be sparse for variable selection. The third component $\mathbf{W}_3 \mathbf{h}_{j_3}$ models other variations such as technical variation. We believe that $\mathbf{h}_{j_1}$ and $\mathbf{h}_{j_2}$ more likely represent the biological variation, and thus we use the learned $\mathbf{h}_{j_1}$ and $\mathbf{h}_{j_2}$ as the low-dimensional representation for cell $j$. The parameter $\theta$ is pre-specified to determine the size of the rare cell types. In practice, we set the parameters $\tau_0 = 0.9$, $\tau_1 = 5$, and $\theta = 0.1$. We set both $K_2$ and $K_3$ equal to 5, and set $K_1$ equal to the number of reference bulk/pseudo-bulk samples, as we used all the principal components (PCs) in the reference data. When more than 30 bulk samples are used to construct the reference, we retained the loading matrix of the first 30 PCs learned from the reference data when implementing RA3. The performance of RA3 is robust to the choice of $\tau_0$, $\tau_1$, $\theta$, $K_2$, and $K_3$ (Supplementary Figs. 13 and 14) and the number of PCs learned from the reference data (Supplementary Fig. 15). In practice, we found that as long as $K_1$ passes certain value, RA3 performs well and stable when it further increases. The reason is that the prior on $\mathbf{h}_{\cdot 1}$ has a shrinkage effect, the variations of $\mathbf{h}_{\cdot 1}$ for the irrelevant projection vectors are small and will be further shrunk toward 0. We set the default value of $K_1$ to be capped at 30. The value of $K_1$ has a minor effect on the computational time of RA3, compared to the scale of the target scCAS dataset (Supplementary Fig. 11b).

**Model fitting and parameter estimation.** Given the observed scCAS matrix $\mathbf{Y} \in \mathbb{R}^{p \times n}$ after TF-IDF transformation and the matrix of covariates $\mathbf{X} \in \mathbb{R}^{q \times n}$, we treat the latent variable matrix $\mathbf{H} = [\mathbf{h}_1, \ldots, \mathbf{h}_n] \in \mathbb{R}^{K \times n}$ as missing data, and use the expectation-maximization (EM) algorithm to estimate the model parameters $\boldsymbol{\Theta} = (\boldsymbol{\Gamma}, \mathbf{A}, \mathbf{W}^*, \boldsymbol{\beta}, \sigma)$, where $\boldsymbol{\Gamma} = [\boldsymbol{\gamma}_1, \ldots, \boldsymbol{\gamma}_n] \in \mathbb{R}^{K \times n}$, $\mathbf{A} = \text{diag}(\alpha_{K_1+1}, \ldots, \alpha_K) \in \mathbb{R}^{(K_2+K_3) \times (K_2+K_3)}$ and $\mathbf{W}^* = [\mathbf{W}_2 \ \mathbf{W}_3] = [\mathbf{w}_{K_1+1}, \ldots, \mathbf{w}_K] \in \mathbb{R}^{p \times (K_2+K_3)}$. For simplicity of the derivation that will follow, we also write $\mathbf{H} = [\mathbf{H}_1^T \ \mathbf{H}_2^T \ \mathbf{H}_3^T]^T$, where $\mathbf{H}_1 \in \mathbb{R}^{K_1 \times n}$, $\mathbf{H}_2 \in \mathbb{R}^{K_2 \times n}$, and $\mathbf{H}_3 \in \mathbb{R}^{K_3 \times n}$, corresponding to the decomposition of variation introduced previously, and $\mathbf{h}_j^* = [\mathbf{h}_{j2}^T \ \mathbf{h}_{j3}^T]^T$, $\mathbf{H}^* = [\mathbf{H}_2^T \ \mathbf{H}_3^T]^T$.

In the expectation step (E-step), given the parameters estimated in the previous iteration $\boldsymbol{\Theta}_{t-1}$, the posterior distribution of $\mathbf{h}_j$ is

$$\mathbf{h}_j | \mathbf{y}_j, \mathbf{x}_j, \boldsymbol{\Theta}_{t-1} \sim \mathcal{N}_K(\hat{\boldsymbol{\mu}}_j, \hat{\boldsymbol{\Sigma}}_j), \tag{6}$$

where $\hat{\boldsymbol{\Sigma}}_j = (\sigma^{-2}\mathbf{W}^{\mathrm{T}}\mathbf{W} + \mathbf{D}_j^{-1})^{-1}$, $\hat{\boldsymbol{\mu}}_j = \sigma^{-2}(\mathbf{y}_j - \boldsymbol{\beta}\mathbf{x}_j)^{\mathrm{T}}\mathbf{W}\hat{\boldsymbol{\Sigma}}_j$, and

$$\mathbf{D}_j = \left[ \mathbf{I}_{K_1} \cdots 0 (1 - \gamma_{(K_1+1)j})\tau_0^2 + \gamma_{(K_1+1)j}\tau_1^2 \cdots 0 \vdots \cdots \vdots 0 \cdots (1 - \gamma_{(K_1+K_2)j})\tau_0^2 + \gamma_{(K_1+K_2)j}\tau_1^2 0 \cdots \mathbf{I}_{K_3} \right] \in \mathbb{R} \quad (7)$$

Based on the posterior distribution, we compute the following expectations:

$$\begin{cases} \mathrm{E}(\mathbf{h}_j) = \hat{\boldsymbol{\mu}}_j^{\mathrm{T}}, \\ \mathrm{E}(\mathbf{h}_j\mathbf{h}_j^{\mathrm{T}}) = \mathrm{E}(\mathbf{h}_j)\mathrm{E}(\mathbf{h}_j)^{\mathrm{T}} + \hat{\boldsymbol{\Sigma}}_j. \end{cases} \quad (8)$$

In the maximization step (M-step), we maximize the expected value of the complete log-likelihood with respect to $\boldsymbol{\Theta}$. The corresponding objective function is

$$\begin{aligned} Q &= \mathrm{E}_{\mathbf{H}|\mathbf{Y},\boldsymbol{\theta}}(\log P(\mathbf{Y}|\mathbf{W},\mathbf{H},\mathbf{X},\mathbf{A},\boldsymbol{\beta}) + \log P(\mathbf{W}|\mathbf{A}) + \log P(\mathbf{H}|\boldsymbol{\Gamma}) + \log P(\boldsymbol{\Gamma})) \\ &= -\frac{np}{2}\log\sigma^2 - \frac{1}{2}\mathrm{E}_{\mathbf{H}|\mathbf{Y},\boldsymbol{\theta}}\left(\mathrm{trace}\left[\sigma^{-2}(\mathbf{Y}-\boldsymbol{\beta}\mathbf{X})^{\mathrm{T}}(\mathbf{Y}-\boldsymbol{\beta}\mathbf{X}) - 2\sigma^{-2}(\mathbf{Y}-\boldsymbol{\beta}\mathbf{X})^{\mathrm{T}}\mathbf{W}\mathbf{H} + \sigma^{-2}(\mathbf{W}\mathbf{H})^{\mathrm{T}}\mathbf{W}\mathbf{H}\right]\right) \\ &\quad + \sum_{k=K_1+1}^{K}\left[-\frac{1}{2}\log|\alpha_k^{-1}\mathbf{I}_p| - \frac{1}{2}\alpha_k\mathbf{w}_k^{\mathrm{T}}\mathbf{w}_k\right] \\ &\quad - \frac{1}{2}\sum_{j=1}^{n}\sum_{k=1}^{K}\left[\mathrm{E}\left(\mathbf{h}_j\mathbf{h}_j^{T}\right)_{kk}^2 \varphi(\gamma_{kj}) - \log\varphi(\gamma_{kj})\right] \\ &\quad + \sum_{j=1}^{n}\sum_{k=K_1+1,\dots,K_1+K_2}\left[\gamma_{kj}\log\theta + (1-\gamma_{kj})\log(1-\theta)\right], \end{aligned} \quad (9)$$

where $\varphi(\gamma_{kj}) = \begin{cases} (1-\gamma_{kj})\tau_0^{-2} + \gamma_{kj}\tau_1^{-2}, & k = K_1+1,\dots,K_1+K_2 \\ 1, & k = 1,\dots,K_1,K_1+K_2+1,\dots,K \end{cases}$, and $\mathrm{E}\left(\mathbf{h}_j\mathbf{h}_j^{T}\right)_{kk}$ is the $k$th diagonal element of matrix $\mathrm{E}(\mathbf{h}_j\mathbf{h}_j^{\mathrm{T}})$.

The optimization problem can be solved by the iterative conditional mode algorithm: in each step, we search for the mode of one group of variables, fixing the other variables. To update $\mathbf{A}$, $\mathbf{W}$, $\boldsymbol{\beta}$ and $\sigma$, we use the following formulas derived by setting the gradients to zero:

$$\alpha_k = \frac{p}{\mathbf{w}_k^{\mathrm{T}}\mathbf{w}_k}, \quad \text{for } k = K_1+1,\dots,K \quad (10)$$

$$\mathbf{W}^* = (\mathbf{Y} - \boldsymbol{\beta}\mathbf{X} - \mathbf{W}_1\mathbf{H}_1)\mathrm{E}(\mathbf{H}^*)^{\mathrm{T}}\left(\sum_{j=1}^{n}\mathrm{E}\left(\mathbf{h}_j^*\mathbf{h}_j^{*\mathrm{T}}\right) + \sigma^2\mathbf{A}\right)^{-1}, \quad (11)$$

$$\boldsymbol{\beta} = (\mathbf{Y} - \mathbf{W}\mathbf{H})\mathbf{X}^{\mathrm{T}}(\mathbf{X}\mathbf{X}^{\mathrm{T}})^{-1}, \quad (12)$$

$$\sigma^2 = \frac{\sum_{j=1}^{n}\left[(\mathbf{y}_j - \boldsymbol{\beta}\mathbf{x}_j)^{\mathrm{T}}(\mathbf{y}_j - \boldsymbol{\beta}\mathbf{x}_j) - 2(\mathbf{y}_j - \boldsymbol{\beta}\mathbf{x}_j)^{\mathrm{T}}\mathbf{W}\mathrm{E}(\mathbf{h}_j) + \mathrm{trace}(\mathrm{E}(\mathbf{h}_j\mathbf{h}_j^{\mathrm{T}})\mathbf{W}^{\mathrm{T}}\mathbf{W})\right]}{pn}. \quad (13)$$

The variable $\boldsymbol{\Gamma}$ is updated element-wise by choosing $\gamma_{kj} \in \{0,1\}$ that maximizes

$$f_{\gamma_{kj}}(\gamma_{kj}) = -\frac{1}{2}\mathrm{E}(\mathbf{h}_j\mathbf{h}_j^{\mathrm{T}})_{kk}^2 \varphi(\gamma_{kj}) + \frac{1}{2}\log\varphi(\gamma_{kj}) + \gamma_{kj}\log\theta + (1-\gamma_{kj})\log(1-\theta), \quad (14)$$

where $\varphi(\gamma_{kj})$ is defined as

$$\begin{cases} \varphi(\gamma_{kj}) = 1, & k = 1,\dots,K_1,K_1+K_2+1,\dots,K \\ \varphi(\gamma_{kj}) = (1-\gamma_{kj})\tau_0^{-2} + \gamma_{kj}\tau_1^{-2}, & k = K_1+1,\dots,K_1+K_2. \end{cases} \quad (15)$$

We iteratively repeat the above E-step and M-step until convergence. $\mathrm{E}(\mathbf{H}_1)$ and $\mathrm{E}(\mathbf{H}_2)$ obtained from the last iteration of the EM algorithm are used for downstream analyses, including data visualization, cell clustering, trajectory inference, and motif analysis. We implement an optional post-processing step for $\mathrm{E}(\mathbf{H}_2)$. We first perform the one-sample $t$-test for each row of the estimate of $\mathrm{E}(\mathbf{H}_2)$ obtained from the final iteration in EM algorithm, and test if its distribution has a zero mean. Here we do not include the multiple testing correction procedure, as the number of rows for testing ($K_2$) is usually small. The rows that cannot reject the null hypothesis under the 0.05 significant level would be discarded for following analysis. We then truncate the values in each remaining row of $\mathrm{E}(\mathbf{H}_2)$ by the 5th and 95th percentile. The purpose of the post-processing step is to alleviate the effect of noisy cells with low sequencing depth. In practice, some components in $\mathbf{H}_2$ may capture very few cells (<5) with extreme values, and these cells are usually noisy cells with low sequencing depth. The optional post-processing step has minimal effect on data visualization but it can improve the clustering result, especially for methods based on Euclidean space, such as $k$-means clustering.

**Initialization.** The initial values for $\boldsymbol{\Theta} = (\boldsymbol{\Gamma}, \mathbf{A}, \mathbf{W}^*, \boldsymbol{\beta}, \sigma)$ are needed for the EM algorithm. We consider a warm start for $\mathbf{W}_2$. More specifically, we calculate the residual matrix $r_{\mathbf{W}_1}$ by subtracting the projection onto the column space of $\mathbf{W}_1$ from $\mathbf{Y}$: $r_{\mathbf{W}_1} = \mathbf{Y} - \mathbf{W}_1\mathbf{W}_1^T\mathbf{Y}$. We then apply PCA to the residual matrix $r_{\mathbf{W}_1}$, and obtain the loading matrix $\mathbf{V}$ and the PC score matrix. Varimax rotation is then implemented on the PC score matrix, and rotation matrix $\mathbf{R}$ is obtained. We then initialize $\mathbf{W}_2$ as $\mathbf{V}\mathbf{R}$. The intuition for varimax rotation is that we are looking for directions within the principal subspace that separates a small number of cells from the rest, mimicking the sparsity prior on $\mathbf{W}_2$. We also consider a warm start for $\sigma$.

To initialize $\sigma$, we first obtain the residual by subtracting from $\mathbf{Y}$ the projection onto the column space of $\mathbf{W}_1$ and $\mathbf{V}\mathbf{R}$, and we then initialize $\sigma$ as the standard deviation of the residual. We initialize $\mathbf{A}$ with the identity matrix, and $\boldsymbol{\beta}$ with zeros. We initialize $\gamma_{kj}$ with 0 for $k = K_1+1,\dots,K_1+K_2$ and 1 for other $k$. $\mathbf{W}_3$ is initialized by a random matrix with elements distributed as standard Gaussian distribution.

## Datasets and data processing

*Datasets.* The human bone marrow dataset contains single-cell chromatin accessibility profiles across ten populations of immunophenotypically defined human hematopoietic cell types from seven donors[29]. The GM/HEK and GM/HL datasets are mixtures of the cell lines GM12878/HEK293T and GM12878/HL-60, correspondingly[5]. The InSilico mixture dataset was constructed by computationally combining scCAS datasets from H1, K562, GM12878, TF-1, HL-60, and BJ cell lines[4]. The mouse forebrain dataset was generated from the forebrain tissue from an 8-week-old adult mouse (postnatal day 56) by single-nucleus ATAC-seq[30]. The MCA mouse brain dataset was generated from the prefrontal cortex, cerebellum, and two samples from the whole brain of 8-week-old mice using sciATAC-seq, which is a combinatorial indexing assay[10]. The 10X PBMC dataset produced by 10X Chromium Single Cell ATAC-seq was generated from PBMC from a healthy donor.

*Data preprocessing.* To reduce the noise level, we selected peaks/regions that have at least one read count in at least 3% of the cells in the scCAS count matrix. Similar to Cusanovich et al.[10], we performed TF-IDF transformation to normalize the scCAS count matrix before implementing our model: we first weighted all the regions in individual cells by the term frequency, which is the total number of accessible regions in that cell, and then multiplied these weighted matrix by the logarithm of inverse document frequency, which is the inverse frequency of each region to be accessible across all cells.

*Downsampling procedure.* We randomly dropped out the non-zero entries in the data matrix to zero with probability equal to the dropout rate.

## Construction of reference

*Manually curated bulk reference.* For the human bone marrow dataset, we used bulk ATAC-seq data of HSC, MPP, LMPP, CMP, GMP, MEP, Mono, CD4, CD8, NK, NKT, B, CLP, Ery, UNK, pDC, and Mega cells provided by Buenrostro et al.[29] as the reference data in RA3. To construct the reference for the GM/HEK, GM/HL, and InSilico mixture datasets, we first downloaded BAM files of bulk DNase-seq samples with relevant biological context from ENCODE[15,16]: GM12878 and HEK293T cell lines for the GM/HEK dataset, GM12878 and HL-60 cell lines for the GM/HL dataset, and H1, K562, GM12878, HL-60, and BJ cell lines for the InSilico mixture dataset. We then counted the reads that fall into the regions of scCAS data for the bulk samples to form a count matrix that has same features/regions as the target single-cell data. We finally obtained the reference data through scaling the count matrix by total mapped reads of each bulk sample.

*OPENANNO.* The webserver OPENANNO[31] provides a convenient and straightforward approach to construct the reference. OPENANNO can annotate chromatin accessibility of arbitrary genomic regions by the normalized number of reads that fall into the regions using BAM files, or the normalized number of peaks that overlap with the regions using BED files. After submitting the peak file in scCAS data to the webserver, OPENANNO computes the accessibility of the single-cell peaks across 199 cell lines, 48 tissues, and 11 systems based on 871 DNase-seq samples from ENCODE. When more than 30 bulk samples are used to construct the reference, we retained the loading matrix of the first 30 PCs learned from the reference data when implementing RA3.

*Pseudo-bulk reference by aggregating single cells.* Given that it can be difficult to obtain the bulk samples for certain cell populations, we can alternatively construct pseudo-bulk reference data by aggregating single cells of the same type/cluster. To address the bias of the difference in cell type abundance in constructing the reference, we took average (instead of taking sum) for each peak over single cells of the same type/cluster. When the peaks do not match between the target single-cell data and the reference single-cell data, we mapped the reads in the reference single-cell data to the peaks in the target single-cell data to obtain the count matrix for the reference single-cell data. The pseudo-bulk reference constructed in this way requires the BAM files for the reference single-cell data. When the BAM files for the reference single-cell data are not available, we can map the reads in the target single-cell data to the peaks in the reference single-cell data instead. We demonstrated this second way of constructing pseudo-bulk reference through the 10X PBMC dataset, and the performance was similar.

## GREAT analysis, trajectory inference, and motif analysis

*GREAT analysis.* We first selected the top 1000 peaks with largest magnitude in the loadings of the second component: we focus on peaks with negative loadings because the sign of $\mathbf{H}_2$ for the identified cell subpopulation (mainly CLP cells) is mostly negative, and the peaks with negative loadings tend to be more accessible in

the identified cell subpopulation. We then submitted these peaks to the GREAT[37] server with a whole genome background and the default parameter settings to identify significant pathways associated with the peaks and thus obtain functional insight on the identified cell subpopulation. We note that our analysis with GREAT does not require knowing the cell labels.

*Trajectory inference.* We adopted Slingshot[52] for trajectory inference, which is the best method of single-cell trajectory inference for tree-shaped trajectories as suggested in a benchmark study[53]. With the default parameter settings, we used the getLineages function to learn cluster relationships using the low-dimensional representation provided by RA3 and the cluster labels obtained from RA3 + Louvain clustering, and then constructed smooth curves representing the estimated cell lineages using the getCurves function. We then used the embedCurves function to map the curves to the two-dimensional t-SNE space for visualization.

*Motif enrichment analysis.* With the cluster labels obtained from RA3 + Louvain clustering, we identified the cluster-specific peaks by implementing the hypothesis testing procedure in scABC[8] on the raw read counts. We selected the top 1000 peaks with smallest *p* values for each cluster, and then performed TF binding motif enrichment within these peaks using chromVAR[7].

**Discussion on the Gaussian assumption**. TF-IDF transformation is necessary for better separation of the cells. After TF-IDF transformation, the data matrix is no longer discrete and becomes continuous. Gaussian distribution is commonly used to model continuous data and it has high computational efficiency given its conjugacy in our model. Computation is an important concern given the scale of the datasets. We used the Kolmogorov–Smirnov (KS) test to test the normality of individual peaks within each cell type after TF-IDF transformation. It is a one-dimensional KS test as testing high-dimensional normality is not feasible. It showed that up to 90% of the one-dimensional tests have been rejected, which suggests that TF-IDF transformed data are not normally distributed. However, given the good performance in all our examples, we think that the Gaussian assumption is robust for cell-level analysis, including visualization, clustering, and lineage reconstruction. For feature-level analysis, including differential peak analysis, the raw read counts should be used. We implemented scABC to detect differential peaks, using the raw read counts and the cluster labels obtained by RA3 + Louvain clustering.

**Baseline methods**. We compared the performance of RA3 with six baseline methods (using their default parameters), including scABC[8], SCALE[13], Scasat[11], cisTopic[9], Cusanovich2018[5,10], and SnapATAC[12]. Source code for implementing the baseline methods was obtained from a benchmark study[6]. We set uniform random seeds in all experiments to ensure the reproducibility of results. Similar to SCALE, we applied all the baseline methods to reduce the scATAC-seq data to ten dimensions for downstream visualization and clustering. To ensure data consistency, we disabled the parameters for filtering peaks and cells in SCALE, and we used the peaks in single-cell data instead of binning the genome into fixed-size windows when implementing SnapATAC.

**Visualization and clustering**
*Visualization.* We first obtained the low-dimensional representation provided by each method. To further reduce the dimension to two, we then implemented t-SNE using the R package Rtsne[34], and UMAP using the R package UMAP[50].

*Clustering.* We first obtained the low-dimensional representation provided by each method except for scABC, and then implemented Louvain clustering[62–64] by scanpy[64], which is a community detection-based clustering method. The cluster assignments for scABC were directly obtained from the model output. We implemented binary search to tune the resolution parameter in Louvain clustering to make the number of clusters and the number of cell types as close as possible[6]. We used eight as the number of expected cell populations for the 10X PBMC dataset.

**Metrics for evaluation of clustering results**. We evaluated the clustering methods based on ARI, NMI, AMI, and homogeneity scores.

Let **T** denote the known ground-truth labels of cells, **P** denote the predicted clustering assignments, $N$ denote the total number of single cells, $x_i$ denote the number of cells assigned to the $i$th cluster of **P**, $y_j$ denote the number of cells that belong to the $j$th unique label of **T**, and $n_{ij}$ denote the number of overlapping cells between the $i$th cluster and the $j$th unique label. Rand index (RI) represents the probability that the obtained clusters and the provided cell type labels will agree on a randomly chosen pair of cells. ARI is an adjusted version of RI, where it adjusts

for the expected agreement by chance, and it is calculated as follows:

$$\mathrm{ARI} = \frac{\sum_{ij}\binom{n_{ij}}{2} - \left[\sum_i \binom{x_i}{2}\sum_j \binom{y_j}{2}\right]/\binom{N}{2}}{\frac{1}{2}\left[\sum_i \binom{x_i}{2} + \sum_j \binom{y_j}{2}\right] - \left[\sum_i \binom{x_i}{2}\sum_j \binom{y_j}{2}\right]/\binom{N}{2}}.$$

Both NMI and AMI are based on mutual information (MI), which assesses the similarity between the obtained clusters and the cell type labels. NMI scales MI to be between 0 and 1, and it is calculated as follows:

$$\mathrm{NMI} = \frac{\mathrm{MI}(\mathbf{P}, \mathbf{T})}{\sqrt{\mathrm{H}(\mathbf{P})\mathrm{H}(\mathbf{T})}},$$

where H(·) is the entropy function.

AMI adjusts MI by considering the expected value under random clustering, and it is calculated as follows:

$$\mathrm{AMI} = \frac{\mathrm{MI}(\mathbf{P}, \mathbf{T}) - \mathrm{E}[\mathrm{MI}(\mathbf{P}, \mathbf{T})]}{\mathrm{avg}[\mathrm{H}(\mathbf{P}), \mathrm{H}(\mathbf{T})] - \mathrm{E}[\mathrm{MI}(\mathbf{P}, \mathbf{T})]},$$

where E(·) denotes the expectation function.

The homogeneity score assesses whether the obtained clusters contain only cells of the same cell type, and it is equal to 1 if all the cells within the same cluster correspond to the same cell type. The homogeneity score is computed as follows:

$$\mathrm{Homogeneity} = 1 - \frac{\mathrm{H}(\mathbf{T}|\mathbf{P})}{\mathrm{H}(\mathbf{T})},$$

where H(**T**|**P**) indicates the uncertainty of true labels based on the knowledge of predicted assignments.

A comparison of ARI, NMI, and AMI was presented in[65]. ARI is preferred when there are large equal-sized clusters[65]. AMI is theoretically preferred to NMI, even though NMI is also very commonly used. AMI is preferred when the sizes of clusters are unbalanced and when there are small clusters[65].

We adopted the RAGI score[6] to evaluate the clustering performance for the 10X PBMC dataset, since no cell type labels are provided in this dataset. The RAGI score calculates the difference between (a) the variability of marker gene accessibility across clusters and (b) the variability of housekeeping gene accessibility across clusters. More specifically, we first used the Gene Scoring method[66] to summarize the accessibility for each gene in each cell. We then computed the mean gene score among cells within each cluster, and computed the Gini index[67] for each marker gene[48] based on these mean gene scores. We use a set of annotated housekeeping genes reported in https://m.tau.ac.il/elieis/HKG/HK_genes.txt. The Gini index for each housekeeping gene is calculated similarly as that for the marker gene. We take average of the Gini indexes within the set of marker genes and the set of housekeeping genes, respectively. Last we obtain the RAGI score by calculating the difference in the average Gini indexes in these two sets of genes. Intuitively, a good clustering result should contain clusters that are enriched for accessibility of the marker genes, and each marker gene should be highly accessible in only one or a few clusters. A larger RAGI score indicates a better separation of the clusters.

**Reporting summary**. Further information on research design is available in the Nature Research Reporting Summary linked to this article.

## Data availability
The human bone marrow dataset and the corresponding reference ATAC-seq data were retrieved from NCBI Gene Expression Omnibus (GEO) with the accession number GSE96772. Single-cell data of the GM/HEK and GM/HL datasets are available at GEO GSE68103. The InSilico mixture dataset was collected from GEO with the accession number GSE65360. BAM files of the reference data for GM/HEK, GM/HL, and InSilico mixture datasets were obtained from ENCODE with the following accession number: ENCFF774HUB, ENCFF775ZJX, and ENCFF783ZLL for the GM/HEK dataset; ENCFF783ZLL, ENCFF775ZJX, ENCFF746RUG, and ENCFF328JJT for the GM/HL dataset; and ENCFF328JJT, ENCFF826DJP, ENCFF923SKV, ENCFF775ZJX, and ENCFF949CIK for the InSilico mixture dataset. The mouse forebrain dataset can be accessed in GEO with the accession number GSE100033. The MCA mouse brain dataset is available at http://atlas.gs.washington.edu/mouse-atac. The 10X PBMC dataset is available at https://support.10xgenomics.com/single-cell-atac/datasets. The reference data of fresh PBMCs were downloaded from GEO with the accession number GSE129785. A reporting summary for this article is available as a Supplementary Information file.

## Code availability
The RA3 R package with a detailed tutorial is freely available at https://github.com/cuhklinlab/RA3 (https://doi.org/10.5281/zenodo.4581063)[68]. The source code for reproduction is available at https://github.com/cuhklinlab/RA3_source (https://doi.org/10.5281/zenodo.4581077).

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

## Acknowledgements

This work is supported by the Chinese University of Hong Kong startup grant 4930181, the Chinese University of Hong Kong direct grant 2018/2019 No. 4053360, direct grant 2019/2020 No. 4053423, Hong Kong Research Grant Council Grants ECS 24301419 and GRF 14301120, the Chinese University of Hong Kong's Project Impact Enhancement Fund (PIEF) and Science Faculty's Collaborative Research Impact Matching Scheme (CRIMS), the National Key Research and Development Program of China (No. 2018YFC0910404), the National Natural Science Foundation of China (Nos. 61873141, 61721003, 61573207), and the Tsinghua-Fuzhou Institute for Data Technology. We thank Kui Hua for his helpful suggestions. We thank Lei Xiong for the cell type labels of the mouse forebrain dataset.

## Author contributions

Z.L. and R.J. conceived and supervised the project. Z.L., S.C., and G.Y. designed, implemented, and validated RA3. W.Z. and J.L. helped analyzing the results. S.C., Z.L., R.J., and G.Y. wrote the manuscript with inputs from all the authors.

## Competing interests

The authors declare no competing interests.
