## [Peer Review File · Nature Communications]

Reviewers' Comments:

Reviewer #1:

Remarks to the Author:

The manuscript "A reference-guided approach for epigenetic characterization of single cells" describes a new computational tool, called "RA3", that utilizes the information in pre-existing bulk chromatin accessibility and annotated single cell data to identify cell types and aid in downstream analysis of single cell chromatin accessibility data. The paper makes four major claims.

(1) RA3 effectively extracts biological variation in single-cell data for downstream

Analyses by incorporating reference data

(2) RA3 also captures the unique biological variation in single cell data that is not represented in the reference data

(3) RA3 can model known covariates, such as donor labels

(4) RA3 consistently outperforms existing methods on datasets generated from different platforms, and of diverse sample sizes and dimensions

Single cell analysis is a very important current topic in biomedical research and overall this manuscript represents a novel contribution to the rapidly expanding literature on this topic. One of the unique aspects of their approach is the ability to use many types of pre-existing data whether it is focused datasets like pseudobulk ATAC data or large curated datasets like ENCODE or CistromeDB. The ability to use either bam files or bed files for the reference data is also an advantage of their approach that is demonstrated in the paper. This manuscript will likely be strong interest to the single cell community and overall the authors do a good job supporting their claims, however, the manuscript would benefit from providing more details in the comparisons with other approaches. Specifically:

(1) The authors compare their results with other approaches using numerous metrics (RAGI, AMI, ARI, NMI, Homogeneity). While this is admirable, it would greatly help the reader to include a discussion of the strengths and weaknesses of these metrics. What exactly are the differences between these metrics that they are reporting? Are the increases being reported for RA3 versus the other methods in figure 4 meaningful?

(2) There is no discussion of how performant their method is. The authors should provide some run time metrics for their analyses and compare the results with some of the other approaches mentioned in the paper. Also how do these metrics scale with the size of the scCAS dataset? With the size of the reference data?

Reviewer #2:

Remarks to the Author:

In this well-written manuscript, the authors introduce a new method RA3 for analyzing single-cell chromatin accessibility sequencing (scCAS) data. Analysis of scCAS data is a timely and important topic given the rapid development and increasing applications of scCAS technology as well as the sparse and noisy nature of the data. The main novelty of RA3 is that it borrows information from existing bulk or pseudo-bulk chromatin data to improve the low-dimensional embedding of single-cell data which in turn improves cell clustering, visualization and trajectory inference. The model used by RA3 can also account for covariates which is another useful feature. Through real data examples, the authors demonstrated the advantage of using RA3 over a number of existing methods. Overall, RA3 can provide a useful new tool for the scCAS community. However, there are several issues that need to be addressed in order to more convincingly demonstrate the value of this new tool.

Major issues:

1. The authors commented that "most reference-guided methods are designed for single-cell

transcriptome data". However, reference-guided methods for single-cell chromatin data are also available (e.g., SCATE, *Genome Biology*, 21:161) but they are not reviewed and discussed in this manuscript. What are the differences between RA3 and the existing reference-guided single-cell chromatin analysis methods?

2. In the RA3 model, the second component captures the direction of variation that separates a small subset of cells from the other cells. What if a large proportion of cells in the scCAS data come from new cell types not represented by the reference data. Will this approach still work? It may be useful to benchmark RA3's performance as the overlap between the scCAS data and reference data gradually decreases.

3. OPENANNO uses peak from single-cell data to extract information from reference data. A potential problem of this approach is that peaks in new rare cell subpopulations cannot always be reliably detected from single-cell data. If those rare cell subtype-specific peaks cannot be detected at the first place, RA3 may not be able to separate those rare cell subpopulations with other cell types in the second component (even with the help from reference data).

4. What is the rationale for applying TF-IDF transformation to the read count data? Is it used to help identify cell-type specific features to improve clustering? Would it affect the downstream analyses such as peak calling since this transformation may penalize peaks that occur in all cells?

5. After the TF-IDF transformation, the data are assumed to have normal distribution. Have the authors checked whether the transformed data are indeed normally distributed? If the data are not normal, will this current model provide robust performance?

6. The model contains many prespecified parameters without fully explaining how their values are chosen. How robust is the model's performance to different choices of these parameters (e.g., tau_0, tau_1, theta, the number of PCs to keep for the reference data, K_3)? It would be useful to show how the model's performance is affected by the parameter choices, and how the impact of parameter choices changes as the proportion of new cells or cell types not captured by the reference data gradually increases.

To the reviewers,

We want to further thank the two reviewers for their time reviewing and for their constructive comments on our manuscript. “A reference-guided approach for epigenetic characterization of single cells”. We have thoroughly revised and enhanced our manuscript based on their comments. In summary, other than the minor changes, we have made the following six major changes in our revised manuscript, with modifications highlighted in red. These changes are summarized in Fig. 4, and Supplementary Figs. 3, 4, 10-15, as well as the main text and the methods section.

(1) We rewrote Section “Metrics for evaluation of clustering results” by including the discussion on the strengths and weaknesses of the clustering metrics. In addition to the indexes of clustering performance, we also included the clustering tables (Supplementary Fig. 10) and evaluated the stability of each method by implementing bootstrap (Fig. 4).

(2) We benchmarked the running time usage on datasets of different sizes. The results are discussed in the main text and shown in Supplementary Fig. 11, and they demonstrate that RA3 is scalable to large datasets.

(3) We tested the robustness of RA3 to the choice of different model parameters. The results are discussed in the main text and shown in Supplementary Figs. 13-15, and they demonstrate that RA3 is robust to the specification of parameters under certain range.

(4) We assessed the performance of RA3 as the overlap between the scCAS data and reference data gradually decreases. The results are shown in Supplementary Fig. 3 and they demonstrate that RA3 is among the best compared to baseline methods for data visualization and cell clustering.

(5) We presented two approaches in Section “RA3 incorporates pseudo-bulk data as reference” to tackle the issue that some peaks may not be reliably detected for the rare cell subpopulations in the target scCAS data. The results are shown in Supplementary Fig. 4 to demonstrate the effectiveness of the approach.

(6) We added clarifications and discussions on term frequency-inverse document frequency (TF-IDF) transformation and the Gaussian assumption in Sections “The model of RA3” and “Discussion on the Gaussian assumption” .

Next, we will clarify the comments raised by the reviewers point by point, with the reviewers’ points in black and our responses in blue.

REVIEWER #1 (REMARKS TO THE AUTHOR)

The manuscript “A reference-guided approach for epigenetic characterization of single cells” describes a new computational tool, called “RA3”, that utilizes the information in pre-existing bulk chromatin accessibility and annotated single cell data to identify cell types and aid in downstream analysis of single cell chromatin accessibility data. The paper makes four major claims.

- (1) RA3 effectively extracts biological variation in single-cell data for downstream analyses by incorporating reference data
- (2) RA3 also captures the unique biological variation in single cell data that is not represented in the reference data
- (3) RA3 can model known covariates, such as donor labels
- (4) RA3 consistently outperforms existing methods on datasets generated from different platforms, and of diverse sample sizes and dimensions

Single cell analysis is a very important current topic in biomedical research and overall this manuscript represents a novel contribution to the rapidly expanding literature on this topic. One of the unique aspects of their approach is the ability to use many types of pre-existing data whether it is focused datasets like pseudobulk ATAC data or large curated datasets like ENCODE or CistromeDB. The ability to use either bam files or bed files for the reference data is also an advantage of their approach that is demonstrated in the paper. This manuscript will likely be strong interest to the single cell community and overall the authors do a good job supporting their claims, however, the manuscript would benefit from providing more details in the comparisons with other approaches. Specifically:

RESPONSE

We thank the reviewer for recognizing the significance of our work and also for the thoughtful comments that have helped us to improve the manuscript.

SPECIFIC COMMENT 1

The authors compare their results with other approaches using numerous metrics (RAGI, AMI, ARI, NMI, Homogeneity). While this is admirable, it would greatly help the reader to include a discussion of the strengths and weaknesses of these metrics. What exactly are the differences between these metrics that they are reporting? Are the increases being reported for RA3 versus the other methods in figure 4 meaningful?

RESPONSE TO SPECIFIC COMMENT 1

Thank you for the constructive comment.

As mentioned by the reviewer, when the cell type labels are given, we evaluated the clustering results based on four widely used metrics [1-3]: adjusted Rand index (ARI), normalized mutual information (NMI), adjusted mutual information (AMI), and homogeneity (H) score. In the following, we present the strengths and weaknesses of these four metrics. A comparison of ARI, NMI, and AMI is presented in [4]. Rand

index (RI) represents the probability that the obtained clusters and the provided cell type labels will agree on a randomly chosen pair of cells. ARI is an adjusted version of RI, where it adjusts for the expected agreement by chance. ARI is preferred when there are large equal sized clusters [4]. Both NMI and AMI are based on mutual information (MI), which assesses the similarity between the obtained clusters and cell type labels. NMI scales MI to be between 0 and 1. AMI adjusts MI by considering the expected value under random clustering. AMI is theoretically preferred to NMI, even though NMI is also very commonly used. AMI is preferred when the sizes of clusters are unbalanced and when there are small clusters [4]. The homogeneity score assesses whether the obtained clusters contain only cells of the same cell type, and it is equal to 1 if all the cells within the same cluster correspond to the same cell type.

For datasets where the cell type labels are not provided, we evaluated the clustering results based on the Residual Average Gini Index (RAGI) score. RAGI calculates the difference between a) the variability of marker gene accessibility across clusters and b) the variability of housekeeping gene accessibility across clusters. Intuitively, a good clustering result should contain clusters that are enriched for accessibility of the marker genes, and each marker gene should be highly accessible in only one or a few clusters.

We have included the discussion of these metrics in Section “Metrics for evaluation of clustering results” in the revised manuscript.

In the manuscript, we have demonstrated that RA3 better extracts the biological variation in the low-dimensional representation, where results of visualization and clustering performance are shown. In the revised manuscript, to demonstrate that the improvement in clustering by RA3 is meaningful, we present the clustering tables (Supplementary Fig. 10), in addition to showing the indexes of clustering performance. Among the five datasets, RA3 consistently achieved a cleaner separation of the cell types, as the clustering counts are more concentrated on the diagonals of the clustering tables.

Supplementary Fig. 10 The clustering tables on various datasets using different methods.

Supplementary Fig. 10 (continue) The clustering tables on various datasets using different methods.

We also evaluated the stability of RA3 by implementing bootstrap on the datasets in Fig. 4. To be more specific, we generated 10 bootstrap samples by random sampling with replacement for each of the dataset. We then performed cell clustering on the bootstrap samples using different methods as described in version 1 of the manuscript. We added the bars representing the standard errors estimated by bootstrap in Fig. 4 in the revised manuscript. RA3 achieved stable and good performance compared to scABC, SCALE, Scasat, cisTopic, Cusanovich2018, and SnapATAC.

SPECIFIC COMMENT 2

There is no discussion of how performant their method is. The authors should provide some run time metrics for their analyses and compare the results with some of the other

approaches mentioned in the paper. Also how do these metrics scale with the size of the scCAS dataset? With the size of the reference data?

RESPONSE TO SPECIFIC COMMENT 2

Thank you for your constructive comment.

In the revision, we have benchmarked the running time usage on datasets of different sizes (five real datasets in version 1 of the manuscript, and three larger datasets with 10K, 20K, 50K cells generated by sampling with replacement on the dataset of MCA mouse prefrontal cortex). All the tests were run on a machine with an Intel Xeon E5-2660 v4 X CPU with 14 cores, 4 GeForce GTX 1080 Ti GPUs and 500GB of RAM on the CentOS 7 operating system.

The major factor that affects the computational time is the size of the target scCAS dataset. For smaller datasets with less than or around 2,000 cells, all methods (scABC, Cusanovich2018, Scasat, cisTopic, SCALE, SnapATAC, and RA3) take less than or around 100s to finish. SCALE, a deep learning-based method, tends to be unstable and can report error message ‘Nan’. scABC and Scasat do not scale well for datasets with more than 20K cells, likely due to massive memory usage. We note that Cusanovich2018 is based on singular value decomposition, which is a relatively simple implementation with existing packages, and RA3 is only slightly slower than Cusanovich2018 when there are 50K cells. Our proposed RA3 scales well especially on larger datasets.

The number of components that we learn from the reference data (K_1) have a minor effect on the computational time of RA3, compared with the scale of the target scCAS dataset. The upper limit for K_1 is the sample size for the reference data. We used the human bone marrow dataset to benchmark the computational time, as it is the scCAS dataset with large reference data from OPENANNO. We implemented PCA on the reference data and used the projection vectors in the first K_1 PCs in RA3. The computational time for RA3 increases slightly when K_1 increases from 25 to 45 (Supplementary Fig. 11b). The results of *t*-SNE visualization are shown in Supplementary Fig. 15, which demonstrate the robustness of RA3 to the choice of number of PCs learned from the reference data.

To summarize, these results indicate that RA3 is scalable to large datasets.

Supplementary Fig. 11 The comparison of computational efficiency and scalability. **a** The computation time of different methods on datasets of various sizes (five real datasets and three datasets simulated by bootstrap on the dataset of MCA mouse prefrontal cortex). For the large dataset with 50,000 cells, scABC did not finish within 1,000 minutes, Scasat exceeds the memory, and SCALE outputs a "Nan" error message. **b** The computation time of RA3 on the human bone marrow dataset using different number of components learned from reference data. The reference data was constructed from BAM files of all the bulk samples in OPENANNO. All the tests were run on a machine with an Intel Xeon E5-2660 v4 X CPU with 14 cores, 4 GeForce GTX 1080 Ti GPUs and 500GB of RAM on the CentOS 7 operating system.

REVIEWER #2 (REMARKS TO THE AUTHOR)

In this well-written manuscript, the authors introduce a new method RA3 for analyzing single-cell chromatin accessibility sequencing (scCAS) data. Analysis of scCAS data is a timely and important topic given the rapid development and increasing applications of scCAS technology as well as the sparse and noisy nature of the data. The main novelty of RA3 is that it borrows information from existing bulk or pseudo-bulk chromatin data to improve the low-dimensional embedding of single-cell data which in turn improves cell clustering, visualization and trajectory inference. The model used by RA3 can also account for covariates which is another useful feature. Through real data examples, the authors demonstrated the advantage of using RA3 over a number of existing methods. Overall, RA3 can provide a useful new tool for the scCAS community. However, there are several issues that need to be addressed in order to more convincingly demonstrate the value of this new tool.

RESPONSE

We appreciate the reviewer for the enthusiastic assessment and thoughtful comments that have helped us to improve the manuscript.

SPECIFIC COMMENT 1

The authors commented that “most reference-guided methods are designed for single-cell transcriptome data”. However, reference-guided methods for single-cell chromatin data are also available (e.g., SCATE, *Genome Biology*, 21:161) but they are not reviewed and discussed in this manuscript. What are the differences between RA3 and the existing reference-guided single-cell chromatin analysis methods?

RESPONSE TO SPECIFIC COMMENT 1

Thank you for your comment and for pointing out the related publication. We note that SCATE was published on 3 July [5], which was after our initial submission on 16 June. We have added SCATE in the introduction section. The goal of SCATE is to reconstruct and recover the “true” chromatin accessibility level for each region in scCAS data, utilizing the information in bulk chromatin accessibility data, which is similar to the goal of imputation methods developed for scRNA-Seq data. The goal of our proposed RA3 is different, where we incorporate reference data (bulk chromatin accessibility data and pseudobulk chromatin accessibility data) to better extract the low-dimensional representation of the cells. The reference data used in SCATE are bulk chromatin accessibility data, and the reference data used in RA3 include both bulk chromatin accessibility data and pseudobulk chromatin accessibility data constructed by aggregating single-cell data. We think that pseudobulk reference data will be preferred when massive single-cell data are generated by the cell atlas consortiums. The information that is transferred from reference data is also different. Since the goal in SCATE is to recover individual entries in the data matrix, the information in bulk chromatin accessibility data that is transferred include the mean and standard deviation of each individual region, and regions that are co-activated

identified from clustering the regions in bulk chromatin accessibility data. The information that is transferred in RA3 includes the projection vectors learned from the reference data. After searching through Google scholar using keywords “reference”, “reference guide”, “single-cell”, “atac”, SCATE is the most related reference-guided approach for scCAS data that we could find. In the search, we found one method for the analysis of scATAC-Seq data, Destin [6]. Destin is based on weighted principal component analysis, where the peaks have different weights based on the distances to transcription start sites (TSSs) and the relative frequency of the peaks in ENCODE data. Although Destin incorporates external information through the weights, we think that how and what external information is utilized is very different in nature from the reference-guided approaches that we are referring to in the manuscript. We also included Destin in the introduction section.

SPECIFIC COMMENT 2

In the RA3 model, the second component captures the direction of variation that separates a small subset of cells from the other cells. What if a large proportion of cells in the scCAS data come from new cell types not represented by the reference data. Will this approach still work? It may be useful to benchmark RA3’s performance as the overlap between the scCAS data and reference data gradually decreases.

RESPONSE TO SPECIFIC COMMENT 2

Thank you for your constructive comment.

Following the reviewer’s suggestion, we constructed an experiment using the mouse forebrain data (it was presented in Fig. 2b in version 1 of the manuscript), where the overlap between scCAS data and the reference data gradually decreases. More specifically, MG and OC are present only in scCAS data and not in the pseudo-bulk reference data, and the other cell types (AC, EX1, EX2, EX3, IN1, and IN2) are shared among the two datasets. We gradually and randomly downsampled the cells of the shared cell types in scCAS data. Before downsampling, MG and OC constitute 18.1% of the total cells in scCAS data. At the end point of downsampling, only 10% cells of the shared cell types are retained, and MG and OC constitute 68.7% of the total cells in scCAS data. We used the default parameter setting to run RA3. As shown in Supplementary Fig. 3a, RA3 successfully separated MG and OC cells even when they become the majority cell types in scCAS data. We also compared the clustering performance. When the overlap between scCAS data and reference data decreases, RA3 is still among the best and the margin between RA3 and the other methods becomes smaller (Supplementary Fig. 3b), which is as expected.

Supplementary Fig. 3 Performance comparison when the overlap between scCAS data and the reference data gradually decreases, using the mouse forebrain dataset (half) and the incomplete reference data (without MG and OC cells). **a** *t*-SNE visualization of the cells in datasets with downsampling ratios of shared cell types varying from 10% to 90%. **b** The clustering performance using datasets with downsampling ratios of shared cell types varying from 10% to 90%.

SPECIFIC COMMENT 3

OPENANNO uses peak from single-cell data to extract information from reference data. A potential problem of this approach is that peaks in new rare cell subpopulations cannot always be reliably detected from single-cell data. If those rare cell subtype-specific peaks cannot be detected at the first place, RA3 may not be able to separate those rare cell subpopulations with other cell types in the second component (even with the help from reference data).

RESPONSE TO SPECIFIC COMMENT 3

Thank you for your constructive and insightful comment.

Since OPENANNO is only one of the options in implementing RA3, we think that the reviewer is referring to the general problem of unreliable peak detection for the rare cell subpopulations. We would like to point out that if the peaks for the rare cell subpopulations cannot be reliably detected, this will not only affect RA3, but also the other methods based on the peaks identified in single-cell data, including scABC, SCALE, Scasat, and cisTopic. In the following, we present two approaches to tackle

the issue when some peaks cannot be reliably detected for the rare cell subpopulations.

The first approach is to use peaks identified in the reference data. We present an example where we use the peaks in the reference data to calculate accessibility for the 10X PBMC scCAS dataset, and the performance is good: RAGI scores are 0.152 (scCAS peaks) vs 0.150 (reference peaks). The cell atlas consortiums will generate massive single-cell data, which can be used as reference. We expect that the peak annotations in reference data that RA3 can take advantage of will become increasingly comprehensive in the future. So the first approach will become appealing to tackle the issue of peak calling for rare subpopulations when reference data becomes more abundant.

The second approach is to iterate between peak calling and clustering with RA3. The peaks may not be reliably detected at the first place. After implementing RA3 and clustering, we may obtain moderate separation of the cell types. Then we redo peak calling within each obtained cell cluster, and reimplement RA3 on these peaks. We expect that iterations between peak calling and clustering with RA3 will improve peak detection and identification of the rare cell subpopulations. As a proof of concept, we implemented this procedure on the mouse forebrain dataset. We first identified cell type-specific peaks by the hypothesis testing procedure in scABC, and held out the top 500 peaks with minimum p -values for the MG and OC cell types respectively (~1,000 peaks held out). Note that MG and OC cells are not present in the reference data. We used the default parameter setting to implement RA3. As shown in Supplementary Fig. 4a, RA3 cannot distinguish MG and OC cells without the MG- and OC-specific peaks. We then performed cell clustering with RA3 + Louvain clustering, called peaks within each obtained cell cluster (we followed the pipeline of peak calling provided by the original study of the mouse forebrain data [7]), and merged these peaks to obtain new features. Using these new features, RA3 successfully identified MG and OC cells (Supplementary Fig. 4b), which demonstrates the effectiveness of the iterative peak calling strategy. We have added an iterative peak calling module to the software of RA3 to facilitate the detection of peaks in rare cell subpopulations. We note that the first approach and the second approach can also be combined, i.e. using both reference peaks and *de novo* peaks in scCAS data.

Supplementary Fig. 4 An example demonstrating the iterative peak calling and clustering strategy for better identification of the rare cell subpopulations in target scCAS data. **a** *t*-SNE visualization of the cells in the mouse forebrain dataset (half) after holding out the MG- and OC-specific peaks. The latent features were obtained from implementing RA3 using incomplete reference data (without MG and OC cells). **b** *t*-SNE visualization of the cells in the mouse forebrain dataset (half) using peaks obtained by the iterative peak calling strategy. The latent features were obtained from implementing RA3 using incomplete reference data (without MG and OC cells).

SPECIFIC COMMENT 4

What is the rationale for applying TF-IDF transformation to the read count data? Is it used to help identify cell-type specific features to improve clustering? Would it affect the downstream analyses such as peak calling since this transformation may penalize peaks that occur in all cells?

RESPONSE TO SPECIFIC COMMENT 4

Thank you for your comment.

We apologize for not making the rationale of applying Term frequency–inverse document frequency (TF-IDF) clear in version 1 of the manuscript. TF-IDF is widely used for text mining. TF-IDF transformation achieves two goals in analyzing scCAS data: 1) it normalizes for sequencing depth; 2) it upweights peaks/regions that do not occur very frequently and downweights prevalent peaks/regions. In our preliminary analysis, we found that this transformation increases the power to separate the cell types, compared to merely normalizing for sequencing depth. We think that the reason why TF-IDF works is because the peaks/regions that are less frequent tend to represent features that distinguish the cell types; the data is near binary, and we need to upweight these peaks to better separate the cell types. To our knowledge, TF-IDF was first implemented in Cusanovich2018 for analyzing scCAS data. The following is a side-by-side comparison between the visualization of PCA (on the raw count matrix) + tSNE, PCA (after normalizing for sequencing depth only) + tSNE and TF-IDF + PCA + tSNE. Note that TF-IDF improves the separation of certain cell types in mouse forebrain and mouse prefrontal cortex. We have added the rationale of TF-IDF transformation in the first paragraph of the Methods section.

Supplementary Fig. 12 The comparison between PCA (on the raw count matrix), PCA (after normalizing for sequencing depth only) and TF-IDF + PCA. **a** *t*-SNE visualizations of the cells in different datasets using different methods. **b** The clustering performance using latent features extracted by different methods, evaluated by Adjusted Mutual Information (AMI).

Note that we only apply TF-IDF transformation for cell-level analysis, including visualization, clustering, and lineage reconstruction. Cell-level analysis is the first step and critical in scCAS analysis, because as long as the cell types are resolved, feature-level analyses, including peak calling within each identified cell type, differential peak analysis, and differential motif analysis become more straightforward. For downstream feature-level analysis, we do not implement TF-IDF transformation and the analysis should be performed on the raw read counts. We

apologize for the confusion and we have added this clarification in Section “RA3 facilitates trajectory inference and motif analysis”.

SPECIFIC COMMENT 5

After the TF-IDF transformation, the data are assumed to have normal distribution. Have the authors checked whether the transformed data are indeed normally distributed? If the data are not normal, will this current model provide robust performance?

RESPONSE TO SPECIFIC COMMENT 5

Thank you for your constructive comment.

As mentioned in the response to comment 4, TF-IDF transformation is necessary for better separation of the cells. After TF-IDF transformation, the data matrix is no longer discrete and becomes continuous. For continuous data, the common distributions include Gaussian distribution and Gamma distribution. If we use Gamma distribution, we lose the computational efficiency because the model is not conjugate. Computation is an important concern given the scale of the dataset.

In an early version of RA3, we implemented and tested the following model on the raw read count data:

$$y_{ij}|\lambda_{ij} \sim \text{Poisson}(s_j\lambda_{ij}), \text{ for } i = 1, \dots, p$$

$$\log \lambda_j = \beta \mathbf{x}_j + \mathbf{W} \mathbf{h}_j$$

where y_{ij} is the read count for cell j and feature/region i , s_j is the offset representing sequencing depth. The specifications of the other variables are the same as those in the current version. The standard EM algorithm is not applicable given the non-conjugacy and intractable E-step. So we implemented an iterative conditional mode algorithm based on the algorithm proposed in [8], which generalizes PCA to the exponential family. However, the real data performance for this model is not good. In addition, it is computationally slow and unstable, which is in part due to the spike-and-slab prior we introduced in \mathbf{h}_{j2} .

We used the Kolmogorov–Smirnov test to test the normality of individual peaks within each cell type after TF-IDF transformation. It is a one-dimensional KS test as testing high-dimensional normality is not feasible. It showed that up to 90% of the one-dimensional tests have been rejected, which suggests that TF-IDF transformed data are not normally distributed. However, given the good performance in all our examples, we think that the Gaussian assumption is robust for cell-level analysis (visualization, clustering, and lineage reconstruction), and it also leads to efficient computation.

In addition, it was shown that the maximum likelihood estimate of probabilistic PCA (with Gaussian assumption) extracts the principal subspace (i.e. the low-dimensional space of PCA) of the datasets [9]. PCA has been implemented for discrete data, including the implementation on genotyping data (where the entries are 0, 1, and 2), where the principal components are widely used as covariates to adjust for population stratification [10]. In a more recent benchmark study for scRNA-Seq data [11], the low-dimensional representation given by PCA was shown to have reasonable and robust overall performance, and good performance in UMI-clustering and trajectory inference (See Figure 5 in [11])

In summary, we think that the Gaussian assumption is reasonable for cell-level analysis in both the aspects of performance and computation. As mentioned in the response to comment 4, for feature-level analysis, we agree that it is necessary to rigorously model the counts of the peaks. The cell-level analysis needs to be performed before the feature-level analysis and if it is done well, the feature-level analysis becomes much easier.

We have added this clarification and discussion in Section “Discussion on the Gaussian assumption” in the revised manuscript.

SPECIFIC COMMENT 6

The model contains many prespecified parameters without fully explaining how their values are chosen. How robust is the model’s performance to different choices of these parameters (e.g., tau_0, tau_1, theta, the number of PCs to keep for the reference data, K_3)? It would be useful to show how the model’s performance is affected by the parameter choices, and how the impact of parameter choices changes as the proportion of new cells or cell types not captured by the reference data gradually increases.

RESPONSE TO SPECIFIC COMMENT 6

Thank you for the constructive comment.

We apologize for not explaining the choice of parameters clearly in version 1 of the manuscript. We used the same set of parameters for diverse datasets in version 1 of the manuscript, which in some sense demonstrates the robustness of parameter specification in RA3.

To further demonstrate the robustness, we first present the result where we implement RA3 using different values of $\tau_0, \tau_1, \theta, K_2$ and K_3 on the following three datasets in the manuscript: human bone marrow dataset, mouse forebrain dataset, and MCA mouse prefrontal cortex dataset. When we varied one parameter, we fixed the other parameters to the default setting. Note that there is some randomness in the result due to the random signs in PCA (used for initialization) and the stochastic randomness in tSNE (used for visualization). When we use the complete reference, where all cell

types in scCAS are included in the reference data, the performance of RA3 is robust to different values of $\tau_0, \tau_1, \theta, K_2$ and K_3 (Supplementary Fig. 13).

Supplementary Fig. 13 The robustness of RA3 with complete reference to the choice of $\tau_0, \tau_1, \theta, K_2$ and K_3 . When we varied one parameter, we fixed the other parameters to the default setting.

Supplementary Fig. 13 (continue) The robustness of RA3 with complete reference to the choice of $\tau_0, \tau_1, \theta, K_2$ and K_3 . When we varied one parameter, we fixed the other parameters to the default setting.

We then present the result for incomplete reference data, where some cell types present in scCAS data are missing in the reference data: in the human bone marrow dataset, bulk ATAC-seq data generated from HSC, MPP, LMPP and CMP cells are used as reference data; in the mouse forebrain dataset and MCA mouse prefrontal cortex dataset, MG and OC cells are not present in the reference data. The incomplete reference data are the same as those presented in version 1 of the manuscript. As shown in Supplementary Fig. 14, RA3 is robust to the choices of parameters ($\tau_0, \tau_1, \theta, K_2$ and K_3) and successfully separated the cell types not represented in the reference data.

Supplementary Fig. 14 The robustness of RA3 with incomplete reference to the choice of τ_0 , τ_1 , θ , K_2 and K_3 . When we varied one parameter, we fixed the other parameters to the default setting.

Supplementary Fig. 14 (continue) The robustness of RA3 with incomplete reference to the choice of $\tau_0, \tau_1, \theta, K_2$ and K_3 . When we varied one parameter, we fixed the other parameters to the default setting.

Last, we varied the value of K_1 , which is the number of PCs used for the reference data. In the previous implementations, we used all the PCs as the sample size for the reference data is not large (the total number of PCs equals the sample size for the reference data). So here we present results for the human bone marrow dataset and the GM/HEK dataset using OPENANNO as the reference, as the sample size in OPENANNO reference is much larger. As shown in Supplementary Fig. 15, RA3 also achieved stable performance using different numbers of PCs learned from the reference data, constructed from either BAM files (Supplementary Fig. 15a) or BED (Supplementary Fig. 15b) files. In practice, we found that as long as K_1 passes certain value, RA3 performs well and stable when K_1 further increases. We think the reason is that the prior on \mathbf{h}_1 has a shrinkage effect, the variations of \mathbf{h}_1 for the irrelevant projection vectors are small and will be further shrunk towards 0. We set the default value of K_1 to be capped at 30.

Supplementary Fig. 15 The robustness of RA3 to the choice of number of PCs learned from the reference data. When we varied the parameter, we fixed other parameters to the default setting. **a** *t*-SNE visualization of the cells in different datasets using latent features obtained from RA3 with reference constructed from BAM files of all the bulk samples in OPENANNO. **b** *t*-SNE visualization of the cells in different datasets using latent features obtained from RA3 with reference constructed from BED files of all the bulk samples in OPENANNO.

Through multiple examples that are just demonstrated, we have shown that RA3 is robust to different parameter settings under certain range. The results that the proportion of new cell types (not captured by the reference data) gradually increases are shown in the response to comment 2, where we have shown that the default parameter setting of RA3 has robust performance when the proportions of the new cell types vary. We did not perform the experiment where both the proportion of new cell types and the default parameters vary, because of the following two reasons. 1) In practice, for a new dataset, we do not know the groundtruth of the proportions of new cell types; We have shown that the default parameters have stable performance when the proportions of new cell types vary in the response to comment 2; Even if changing the default parameters may lead to a slight improvement, it will be hard to justify because we are not certain on the groundtruth. 2) The number of figures will explode and easily exceed ten pages if we consider different combinations of parameters and proportions of new cell types; We think that the results we have shown have already demonstrated the point.

References

- [1] Chen, H. et al. Assessment of computational methods for the analysis of single-cell atac-seq data. *Genome Biology* **20**, 241 (2019).
- [2] Bravo Gonzalez-Blas, C. et al. cistopic: cis-regulatory topic modeling on single-cell atac-seq data. *Nature Methods* **16**, 397–400 (2019).
- [3] Xiong, L. et al. Scale method for single-cell atac-seq analysis via latent feature extraction. *Nature Communications* **10**, 4576 (2019).
- [4] Romano, S. et al. Adjusting for chance clustering comparison measures. *The Journal of Machine Learning Research* **17**(1), 4635-4666 (2016).
- [5] Ji, Z. et al. Single-cell ATAC-seq signal extraction and enhancement with SCATE. *Genome Biology* **21**(1), 1-36 (2020).
- [6] Urrutia, E., Chen, L., Zhou, H. & Jiang, Y. Destin: toolkit for single-cell analysis of chromatin accessibility. *Bioinformatics* **35**, 3818–3820 (2019).
- [7] Preissl, S. et al. Single-nucleus analysis of accessible chromatin in developing mouse forebrain reveals cell-type- specific transcriptional regulation. *Nature Neuroscience* **21**, 432–439 (2018).
- [8] Collins, M. et al. A generalization of principal components analysis to the exponential family. *Advances in neural information processing systems* (2002).
- [9] Tipping, M. E., & Bishop, C. M. Probabilistic principal component analysis. *Journal of the Royal Statistical Society: Series B (Statistical Methodology)* **61**(3), 611-622 (1999).
- [10] Price, A. L. et al. Principal components analysis corrects for stratification in genome-wide association studies. *Nature Genetics* **38**(8), 904-909 (2006).
- [11] Sun, S. et al. Accuracy, robustness and scalability of dimensionality reduction methods for single-cell RNA-seq analysis. *Genome Biology* **20**(1), 269 (2019).

Reviewers' Comments:

Reviewer #1:

None

Reviewer #2:

Remarks to the Author:

The authors have done a nice job responding to my comments and questions. All my concerns are addressed in the revised manuscript.

REVIEWER #2 (REMARKS TO THE AUTHOR)

The authors have done a nice job responding to my comments and questions. All my concerns are addressed in the revised manuscript.

RESPONSE

We appreciate the reviewer for the enthusiastic assessment that has helped us to improve the manuscript.